



# Thirty-eight years of $CO_2$ fertilization have outpaced growing aridity to drive greening of Australian woody ecosystems

Sami W. Rifai[1, *], Martin G. De Kauwe[1,2,3,4], Anna M. Ukkola[1,5], Lucas A. Cernusak[6], Patrick Meir[7,8], Belinda E. Medlyn[9], and Andy J. Pitman[1,2]

[1]ARC Centre of Excellence for Climate Extremes, University of New South Wales, Sydney, NSW 2052, Australia
[2]Climate Change Research Centre, University of New South Wales, Sydney, NSW 2052, Australia
[3]Evolution & Ecology Research Centre, University of New South Wales, Sydney, NSW 2052, Australia
[4]School of Biological Sciences, University of Bristol, Bristol, BS8 1TQ, United Kingdom
[5]Research School of Earth Sciences, Australian National University, Canberra, ACT 0200, Australia
[6]College of Science and Engineering, James Cook University, Cairns, QLD 4188, Australia
[7]Research School of Biology, The Australian National University, Acton, ACT 2601, Australia
[8]School of Geosciences, University of Edinburgh, Edinburgh EH89XP, UK
[9]Hawkesbury Institute for the Environment, Western Sydney University, Penrith, NSW 2753, Australia

**Correspondence:** Sami W. Rifai (s.rifai@unsw.edu.au)

**Abstract.** Climate change is projected to increase the imbalance between the supply (precipitation) and atmospheric demand for water (i.e. increased potential evapotranspiration), stressing plants in water-limited environments. Plants may be able to offset increasing aridity because rising $CO_2$ increases water-use-efficiency. $CO_2$ fertilization has also been cited as one of the drivers of the widespread 'greening' phenomenon. However, attributing the size of this $CO_2$ fertilization effect is complicated, due in part to a lack of long-term vegetation monitoring and interannual to decadal-scale climate variability. In this study we asked the question, how much has $CO_2$ contributed towards greening? We focused our analysis on a broad aridity gradient spanning eastern Australia's woody ecosystems. Next we analysed 38-years of satellite remote sensing estimates of vegetation greenness (normalized difference vegetation index, NDVI) to examine the role of $CO_2$ in ameliorating climate change impacts. Multiple statistical techniques were applied to separate the $CO_2$-attributable effects on greening from the changes in water supply and atmospheric aridity. Widespread vegetation greening occurred despite a warming climate, increases in vapor pressure deficit, and repeated record-breaking droughts and heatwaves. Between 1982-2019 we found that NDVI increased (median 11.3%) across 90.5% of the woody regions. After masking disturbance effects (e.g. fire), we statistically estimated an 11.7% increase in NDVI attributable to $CO_2$, broadly consistent with a hypothesized theoretical expectation of an 8.6% increase in water-use-efficiency due to rising $CO_2$. In contrast to reports of a weakening $CO_2$ fertilization effect, we found no consistent temporal change in the $CO_2$ effect. We conclude rising $CO_2$ has mitigated the effects of increasing aridity, repeated record-breaking droughts, and record-breaking heat waves in eastern Australia. However, we were unable to determine whether trees or grasses were the primary beneficiary of the $CO_2$ induced change in water-use-efficiency, which has implications for projecting future ecosystem resilience. A more complete understanding of how $CO_2$ induced changes in water-use-efficiency affect trees and non-tree vegetation is needed.



## 1 Introduction

Australia is the world's driest inhabited continent. Predicting how climate change will affect ecosystem resilience and alter Australia's terrestrial hydrological cycle is of paramount importance. Australia's woody ecosystems are mostly concentrated in the east, where there are large gradients of precipitation (P) (300 - 2000+ mm yr$^{-1}$) and potential evapotranspiration (PET) (800-2100 mm yr$^{-1}$). Most eastern Australian woodlands occupy water-limited regions where annual PET far exceeds P (Fig.

A1), and tree species have evolved to cope with water-limited conditions (Peters et al.) and high interannual rainfall variability. However, the climate is warming: eight of Australia's ten warmest years on record have occurred since 2005 (CSIRO and Bureau of Meteorology, 2020) and Australia's climate has warmed by ~1.5°C since records began in 1910. The warming has likely increased atmospheric demand for water (e.g. PET or vapor pressure deficit, VPD). In most woody ecosystems, the ratio of water supply (i.e. P) to water demand (i.e. PET) has declined in recent decades (Figs. 1,2a). Eastern Australia has also

been impacted by several multi-year droughts, episodic deluges of rainfall (King et al., 2020), and an increasing frequency of severe heatwaves (Perkins et al., 2012) in the last few decades. Precipitation changes have been spatially variable over eastern Australia, where northern Queensland grew wetter and southeast Australia grew drier (Fig. A2). In the last two decades, southeast Australia experienced the two worst droughts in the observational record (2001-2009; van Dijk et al. (2013) and 2017-2019; (Bureau of Meteorology, 2019). Yet between these two droughts, Eastern Australia experienced record breaking

rainfall in 2011 associated with a strong La Niña event. This caused marked vegetation 'greening' (e.g. increased foliar cover), even in the arid interior (Bastos et al., 2013; Poulter et al., 2014; Ahlstrom et al., 2015). However, this greening contributed to record-breaking fires in the following year (Harris et al., 2018).

Theory suggests that plant physiological responses to atmospheric carbon dioxide ($CO_2$) may mitigate some of the negative effects of an aridifying climate. However, the magnitude of plant responses to increased atmospheric $CO_2$ has been challenging

to establish in field experiments (Jiang et al., 2020b), from observations (Zhu et al., 2016; Walker et al., 2020), or to separate from other drivers (e.g. climate variability, disturbances, and changes in land management). Studies have used data from the Advanced Very High Resolution Radiometer (AVHRR) satellites to show positive trends in the normalized difference vegetation index (NDVI) over Australia (Donohue et al., 2009). The greening trend is caused by increased leaf area, which has resulted from increased atmospheric $CO_2$ concentrations (Donohue et al., 2009; Ukkola et al., 2016). The evidence for

increases in leaf area from rising $CO_2$ have also been supported by observations of reduced runoff in Australia's drainage basins (Trancoso et al., 2017; Ukkola et al., 2016).

Yet disentangling the $CO_2$ fertilization effect from other drivers of climate variability and global change has been particularly challenging for satellite based analyses. It is challenging to attribute causes of greening because of co-occurring changes in climate, land-use, and disturbance are confounded with the effect of $CO_2$ fertilization. Furthermore, the time series of even the

longest systematically collected optical vegetation index records from a single sensor is 20 years (e.g. MODIS Terra). Analysis of trends extending beyond 20 years requires merging satellite records across sensors and platforms. But this requires care to address changes in radiometric and spatial resolution of the sensor, as well as drift in the solar zenith angle (Ji and Brown, 2017; Frankenberg et al., 2021) and the time of retrieval. Thus different analytical methodologies have produced disagreements





over where greening has occurred (Cortés et al., 2021). One often-used method to provide additional constraint on greening
trends has been to compare remote sensing derived trends with modeled changes in leaf area index (LAI) from ensembles of
dynamic global models (Zhu et al., 2016; Wang et al., 2020). However these model attribution approaches rely on a set of
key assumptions. None of the models can accurately predict LAI changes in response to rising $CO_2$ (De Kauwe et al., 2014;
Medlyn et al., 2016). Vegetation models have been shown to diverge in their simulation of LAI over Australia (Medlyn et al.,
2016; Teckentrup et al., 2021; Zhu et al., 2016), and have bioclimatic rules for determining phenology which may not be
appropriate for the highly variable Australian climate and the evergreen Eucalyptus forests (Teckentrup et al., 2021). These
model simulations are typically compared with modeled LAI products derived from the red and near infrared wavelengths of
multispectral satellite sensors, of which each product carries specific algorithmic assumptions about canopy-light interception
which are conditional upon estimated land cover types. In comparison, NDVI carries no ecosystem specific assumption, and is
an effective proxy of leaf area in ecosystems with low-to-moderate canopy cover (Carlson and Ripley, 1997), a characteristic
of eastern Australian woody ecosystems (Specht, 1972; Yang et al., 2018).

Here we ask, how much can greening trends be explained by rising $CO_2$? Using eastern Australia as a model system, we
used a multi satellite derived NDVI record encompassing 38 years to isolate the influence of $CO_2$ from simultaneous effects
of meteorological change and disturbance. Next we contrasted $CO_2$ effects with theoretical predictions based on water-use-
efficiency (WUE) theory for plants and the observed rise in $CO_2$. Finally, we examined whether recent NDVI greening trends
have co-occurred with changes in tree or grass cover over the last two decades.

## 2  Methods

### 2.1  Study area

The study region encompasses the dominant woody ecosystems of eastern Australia (Fig. A1b). We used the National Veg-
etation Information System 5.1 land cover dataset (Dep; Table A1) to select locations designated as "Acacia Forests and
Woodlands", "Acacia Open Woodlands", "Callitris Forests and Woodlands", "Casuarina Forests and Woodlands", "Eucalypt
Low Open Forests", "Eucalypt Open Forests", "Eucalypt Open Woodlands", "Eucalypt Tall Open Forests", "Eucalypt Wood-
lands", "Low Closed Forests and Tall Closed Shrublands", "Mallee Open Woodlands and Sparse Mallee Shrublands", "Mallee
Woodlands and Shrublands", "Melaleuca Forests and Woodlands", "Other Forests and Woodlands", "Other Open Woodlands",
"Rainforests and Vine Thickets", and "Tropical Eucalypt Woodlands/Grasslands".

### 2.2  Climate and remote sensing datasets

We used the atmospheric $CO_2$ record from the deseasonalized Mauna Loa record (https://www.esrl.noaa.gov/gmd/ccgg/trends/
data.html), and extracted climate data (Table A1) from the Australian Bureau of Meteorology's Australian Water Availabil-
ity Project (AWAP; Jones et al. (2009)). AWAP is a gridded climate product interpolated to 0.05° from a large network of
meteorological stations distributed across Australia. Vapor pressure deficit was calculated using daily estimates of maximum





temperature and vapor pressure at 15:00 hours. PET was calculated from shortwave radiation and mean air temperature using
the Priestley-Taylor method (Davis et al., 2017). The Priestley-Taylor method has been shown to be appropriate for estimating
large-scale PET (Raupach, 2000) and is more suited for use in long-term analysis where $CO_2$ increased than other common
formulations such as the Penman-Monteith equation (Greve et al., 2019; Milly and Dunne, 2016), which explicitly imposes
a fixed stomatal resistance that is incompatible with plant physiology theory (Medlyn et al., 2001). AWAP measurements of

shortwave radiation only extend back to 1990, so we extended the PET record to 1982 by calibrating the ERA5-Land PET
record (1980-2019) to the AWAP PET record (1990-2019) by linear regression for each grid-cell, and then gap-filled the years
1982-1989 with the calibrated ERA5 PET. PET from the Climate Research Unit record (Harris et al., 2014) was highly corre-
lated with both the recalibrated ERA5 PET (r = 0.91; 1982-1989) and the original AWAP PET (r = 0.97; 1990-2019). Next, we
calculated a 30-year climatology of the meteorological variables using the period of 1982-2011 to be close to current standards

(Organization, 2017). We used this climatology to define the mean annual P:PET ($MI_{MA}$), and as the reference to calculate a
12-month running anomaly of annual P:PET ($MI_{anom}$). Zonal statistics for each meteorological variable were calculated using
simplified Köppen climate zones, derived from the Australian Bureau of Meteorology (Fig. 2b, Table A1).

We used surface reflectance from two satellite products to generate the NDVI record: National Oceanic and Atmospheric Ad-
ministration's Climate Data Record v5 Advanced Very High Radiometric Resolution (AVHRR) Surface Reflectance (NOAA-

CDR) record and the National Space and Aeronautical Administration's MCD43A4 Nadir Bidirectional Reflectance Distri-
bution Function Adjusted Reflectance (MODIS-MCD43) (Table A1; Schaaf & Wang, 2015). NDVI data was extracted from
1982-2019 at 0.05° resolution from the NOAA-CDR AVHRR version 5 product (Vermote and NOAA CDR Program, 2018).
The surface reflectance record of AVHRR extends through 2019, but the quality of the record starts to degrade in 2017 because
of an increase in the solar zenith angle (Ji and Brown, 2017), causing a sensor-produced decline in NDVI during 2017-2019.

For this reason we only use AVHRR surface reflectance data between 1982-2016. We composited monthly mean AVHRR
NDVI ($NDVI_{AVHRR}$) estimates using only daily pixel retrievals with no detected cloud cover (Quality Assurance band, bit
1). Monthly $NDVI_{AVHRR}$ estimates aggregated from less than three daily retrievals were removed. They were also removed
when the coefficient of variation of daily retrievals for a given month was greater than 25%. We also removed $NDVI_{AVHRR}$
monthly estimates where $NDVI_{AVHRR}$, solar zenith angle, or time of acquisition deviated beyond 3.5 standard deviations from

the monthly mean, calculated from a climatology spanning 1982-2016.

We used the MODIS-MCD43 surface reflectance at 500 m resolution to derive NDVI for 2001-2019 ($NDVI_{MODIS}$). Monthly
mean estimates of the surface reflectance were produced by compositing pixels flagged as "ideal-quality" (Quality Assurance,
bits 0-1). We also masked disturbances to have greater confidence in our attribution of the targeted drivers of $NDVI_{MODIS}$
change (climate & $CO_2$). The Global Forest Change product v1.7 (Hansen et al., 2013) was used to mask pixels from 2001

onwards that had experienced forest loss due to deforestation or severe stand clearing disturbance. We masked pixel locations
that experienced bushfires from the year 2001 onwards. Specifically, these pixels were masked for the year of burning and
the following three years using the 500 m resolution MODIS-MCD64 monthly burned-area product (Giglio et al., 2018). We
terminated the $NDVI_{MODIS}$ time series in August of 2019, prior to the widespread bushfires of late 2019/2020. Both $NDVI_{AVHRR}$
and $NDVI_{MODIS}$ datasets were processed using Google Earth Engine (Gorelick et al., 2017), and exported at 5 km spatial





resolution, which best approximated the native resolution of the NOAA-CDR AVHRR and AWAP products. Further post-processing used the 'stars' (Pebesma, 2020) and 'data.table' (Dowle and Srinivasan, 2019) R packages (see code availability section).

We merged the processed 1982-2016 NDVI$_{AVHRR}$ with the 2001-2019 NDVI$_{MODIS}$ by recalibrating the NDVI$_{AVHRR}$ with a generalized additive model (GAM). Specifically, we used one million observations from the overlapping 2001-2016 portion of

both records to fit a GAM using the 'mgcv' R package (Wood, 2017) to model NDVI$_{MODIS}$ from AVHRR derived covariates as:

$$NDVI_{MODIS} = s(NDVI_{AVHRR}) + s(month) + s(SZA) + s(TOD) + s(x, y) \qquad (1)$$

where 's' represents a penalized smoothing function using a thin-plate regression spline, $SZA$ is the solar zenith angle, $NDVI_{AVHRR}$ is the uncalibrated NDVI from AVHRR, $TOD$ is time of day of retrieval, and $x$ and $y$ represent longitude and

latitude, respectively. The fit GAM was then used to generate the recalibrated AVHRR NDVI. The merged NDVI dataset was created by joining the 1982-2000 recalibrated AVHRR NDVI with the 2001-2019 NDVI$_{MODIS}$. We further reduced monthly temporal variability of NDVI by calculating a three month rolling mean of NDVI which we used for subsequent statistical model fitting.

### 2.3 Estimating NDVI and climate trends

We estimated the relative increase in NDVI between 1982-2019 with respect to time (equation 2) for each grid cell with an iteratively weighted least squares robust linear model via the 'rlm' function in R's MASS package (Venables and Ripley, 2002) as follows.

$$NDVI = \beta_0 + \beta_1 \, year + \beta_2 \, sensor \qquad (2)$$

Here $\beta_0$ represents the estimated NDVI in 1982, the year term starts at 1982, and the sensor term is a binary covariate that

accounts for residual offset differences between the recalibrated AVHRR NDVI , and the NDVI$_{MODIS}$. The relative temporal trends for climate variables and the MODIS vegetation continuous fractions were fit for each grid cell location using the Theil-Sen estimator, a form of robust pairwise regression, with the 'zyp' R package (Bronaugh and Consortium, 2019). The temporal covariate was recentered to start with the first hydrological year (where the year starts one month earlier in December) of the data so that the intercept term represents the mean at the start of the time series. The relative rate of change for each variable

was reconstructed by calculating

$$100 * \left[ \frac{\beta_1(year_{end} - year_{start})}{\beta_0} \right] \qquad (3)$$

where $\beta_0$ and $\beta_1$ are the intercept and trend derived from Theil-Sen regression.

### 2.4 Estimating contribution of CO$_2$ and climate toward NDVI trends

We used the merged NDVI observations to fit multiple statistical models to quantify the impact of changes in CO$_2$ and meteo-

rological variables on NDVI. The relationship between NDVI and the running 12-month mean of P:PET was strongly nonlinear





and followed a monotonic saturating sigmoidal relationship as indicated by GAM fits (methods equation 6, see below). GAMs can characterize a nonlinear response without specifying a functional form, yet the underlying spline parameters are not easily interpreted as the parameters of a fixed nonlinear function. Therefore we used nonlinear least squares (nls.multstart package (Padfield and Matheson, 2020) in R v4.01) to compare model fits to a set of fixed nonlinear functional forms including the

Weibull function (equation 4; Fig. 4), the logistic function (equation 5; Fig. A5), and the Richards growth function (equation 6; Fig. A6). The focus on the Weibull models because they showed equivalent goodness of fit with fewer parameters than the Richards function models. Next we added a linear modifier to the Weibull function using the covariates of $CO_2$ (ppm) and the ratio of the anomaly of P:PET ($MI_{anom}$) to the mean annual P:PET ($MI_{MA}$) as follows:

$$NDVI = V_a - V_d[exp(-exp(c_{ln})(MI_{MA})^q)] + \eta \tag{4}$$

$$\eta = \beta_1 \frac{MI_{anom}}{MI_{MA}} + \beta_2 CO_2 MI_{MA} + \beta_3 CO_2 \frac{MI_{anom}}{MI_{MA}} + sensor$$

Here the sensor term is a binary covariate indicating the AVHRR or MODIS platform. Model-fitted parameters $V_a$ and $V_d$ correspond to the asymptote, and the asymptote's difference from the minimum NDVI, while $c_{ln}$ is the logarithm of the rate constant, and $q$ is the power to which $MI_{MA}$ is raised. The model was fit by individual season with one million observations per model fit. Corresponding goodness-of fit metrics ($R^2$ and root mean square error) were calculated by season (Fig. 5) with one

million randomly sampled observations. Alternative nonlinear functional forms were also fit to characterize the effect of $CO_2$ upon NDVI. A logistic model was fit across space for each hydrological year as

$$NDVI = \frac{V_A}{(1 + exp((m - MI_{12mo})/s))} \tag{5}$$

where $NDVI$ is the hydrological year mean value of NDVI for a grid cell location, $m$ is the midpoint, $s$ is a scale parameter, and $V_A$ is the asymptote (plotted in Fig. A5). We also used a modified Richards growth function to characterize the $CO_2$ effect

upon seasonal NDVI (Fig. A6) as

$$NDVI = (V_A + \beta_1 CO_2 + \beta_2 MI_{f.anom}) \frac{(1 + exp(m + \beta_3 CO_2 + \beta_4 MI_{f.anom} - MI_{MA}))}{(s + \beta_5 CO_2 + \beta_6 MI_{f.anom})^{(-exp(-(q + \beta_7 CO_2 + \beta_8 F)))}} \tag{6}$$

$$MI_{f.anom} = \frac{MI_{anom}}{MI_{MA}}$$

Here the $\beta$ terms act to linearly modify the core nonlinear parameters ($V_A, m, s, q$) with the effects of $CO_2$ and $MI_{f.anom}$. Each seasonal model component was fit across space with one million random samples from the total merged NDVI record

(approx 14.3 million observations).

To ensure consistent interpretation of the nonlinear response across P:PET, we also fit linear models explaining NDVI with $CO_2$ and $MI_{anom}$ by season in $MI_{MA}$ bin-widths of 0.2 (equation 7; Fig. A4). Separate linear models were fit for increments of 0.15 of $MI_{MA}$ for each season using the merged 1982-2019 NDVI record. NDVI was modeled as

$$NDVI = \beta_0 + \beta_1 CO_2 + \beta_2 MI_{anom} + \beta_3 Veg.Class + \beta_4 sensor \tag{7}$$





where MI$_{anom}$ is the annual anomaly of P:PET, Veg. Class is the NVIS 5.1 vegetation class, and $sensor$ is a binary variable used to account for residual differences between the recalibrated AVHRR NDVI and NDVI$_{MODIS}$ records. To aid the comparison of model effects, we centered and standardized the continuous model before regression. The standardized $CO_2$ and P:PET$_{anom}$. effects ($\beta$) are presented in Fig. A4.

Next, we fit robust multiple linear regression models to the time series of NDVI for each of the 39,463 pixel locations. The

$CO_2$ effect for each grid cell location was simultaneously estimated with the linear effects of the anomalies (anom) of P, PET, VPD, and MI as fractions of their mean annual values (MA) as follows.

$$NDVI = \beta_0 + \beta_1\,CO_2 + \beta_2\,\frac{P_{anom}}{P_{MA}} + \beta_3\,\frac{PET_{anom}}{PET_{MA}} + \beta_4\,\frac{VPD_{anom}}{VPD_{MA}} + \beta_5\,sensor \tag{8}$$

Finally we estimated the $CO_2$ effect across the study region using a GAM with a penalized smoothing function ($s$) characterizing the effect of the anomalies and mean annual values of VPD, P, and PET and sensor epoch as follows.

$$NDVI = s(MI_{MA}, CO_2) + s(VPD_{anom}, VPD_{MA}) + s(P_{anom}, P_{MA}) + s(PET_{anom}, PET_{MA}) + sensor \tag{9}$$

### 2.5  A simplified theoretical water use efficiency model

We compared the statistically attributed $CO_2$ amplification of NDVI with the expectation from a simple theoretical model of WUE. Following Donohue et al. (2013), WUE ($W$) is defined as:

$$W_{leaf} = \frac{A_{leaf}}{E_{leaf}} = \frac{C_a}{1.6D}(1 - \chi) \tag{10}$$

where $A$ is leaf level assimilation ($umol\,m^2\,s^{-1}$), E is leaf level transpiration ($mol\,m^2\,s^{-1}$), $C_a$ is atmospheric $CO_2$ ($umol\,umol^{-1}$), $C_i$ is intercellular $CO_2$ ($umol\,umol^{-1}$), $\chi$ is $\frac{C_i}{C_a}$, and $D$ is atmospheric vapor pressure deficit ($mol\,mol^{-1}$). The relative rate of change in $W$ with respect to a change in $C_a$ can be calculated as:

$$\frac{dW_{leaf}}{W_{leaf}} = \frac{dA_{leaf}}{A_{leaf}} - \frac{dE_{leaf}}{E_{leaf}} = \frac{dC_a}{C_a} - \frac{dD}{D} + \frac{d(1 - \chi)}{(1 - \chi)} \tag{11}$$

If temperature increases without a corresponding increase in humidity, $D$ increases which also causes transpiration to rise

and thus reduces $W$. However, $W$ is predicted to increase with $CO_2$ which may offset increases in $D$. Experiments suggest that $\chi$ does not change with $C_a$ but is sensitive to $D$ (Wong et al., 1985; Drake et al., 1997) and can be estimated as being proportional to the square root of $D$ (Medlyn et al., 2011). By substituting

$$(1 - \chi) \approx \sqrt{(D)}$$

into equation (11) we can estimate the theoretical combined effect of $C_a$ and $D$ upon $W_{leaf}$ as:

$$\frac{dW_{leaf}}{W_{leaf}} = \frac{dA_{leaf}}{A_{leaf}} - \frac{dE_{leaf}}{E_{leaf}} = \frac{dC_a}{C_a} - \frac{1}{2}\frac{dD}{D} \tag{12}$$



Transpiration per unit ground area is strongly controlled by water supply in warm, water limited environments with relatively low leaf area such as eastern Australia (Specht, 1972) therefore we approximate canopy transpiration ($E_{canopy}$) as:

$$E_{canopy} = E_{leaf}\, L \tag{13}$$

The change in $E_{canopy}$ can then be defined as:

$$\frac{dE_{canopy}}{E_{canopy}} \approx \frac{dE_{leaf}}{E_{leaf}} + \frac{dL}{L} \tag{14}$$

If we assume there is no long-term overall change in precipitation then we can assume change in $E_{canopy}$ is tightly coupled to the water supply, therefore we have:

$$-\frac{dE_{leaf}}{E_{leaf}} \approx \frac{dL}{L} \tag{15}$$

NDVI is linearly related to foliar cover ($F$) until LAI $\approx 3\ (m^2\, m^{-2})$ (Carlson and Ripley, 1997), which is the predominantly the case when P:PET < 1. Most woody ecosystems of eastern Australia are strongly water limited with LAI $\leq 1\ (m^2\, m^{-2})$, where NDVI is approximately proportional with the fraction of foliar cover:

$$\frac{dL}{L} \approx \frac{dF}{F} \approx \frac{dNDVI}{NDVI} \tag{16}$$

Then substituting equation (15) into equation (12) gives:

$$\frac{dW_{leaf}}{W_{leaf}} \approx \frac{dA_{leaf}}{A_{leaf}} + \frac{dF}{F} \approx \frac{dC_a}{C_a} - \frac{1}{2}\frac{dD}{D} \tag{17}$$

If we assume that the benefit towards $W_{leaf}$ from rising $C_a$ is split evenly between the relative changes in $A_{leaf}$ and $F$, we can predict the change towards NDVI to be

$$\frac{dNDVI}{NDVI} \approx \frac{1}{2}\Big[\frac{dCa}{Ca} - \frac{dD}{2D}\Big] \tag{18}$$

We compared the WUE theoretical model with the robust linear models fit for each pixel location (equation 8), and the GAM (equation 9) fit across the study region. The WUE theoretical model assumes no change in P, but does account for changes in VPD. Therefore in using the statistical models to compare with the WUE predictions, we generated counterfactual predictions from the statistical models with no precipitation anomaly but with the observed increases in $CO_2$ and VPD. One weakness with the application of this WUE theoretical model is the uncertainty regarding the assumed allocation of the $W_{leaf}$ benefit towards either $A_{leaf}$ or $F$ (e.g. LAI; see above). (Donohue et al., 2017) proposed a similar model to eq (18), the Partitioning of Equilibrium Transpiration and Assimilation (PETA) hypothesis where the relative allocation to leaf area is predicted to decline with increasing resource availability (which could be inferred from growing season LAI). We calculated the expectation from the PETA hypothesis as another point of comparison with the $CO_2$ attributable effect on NDVI.



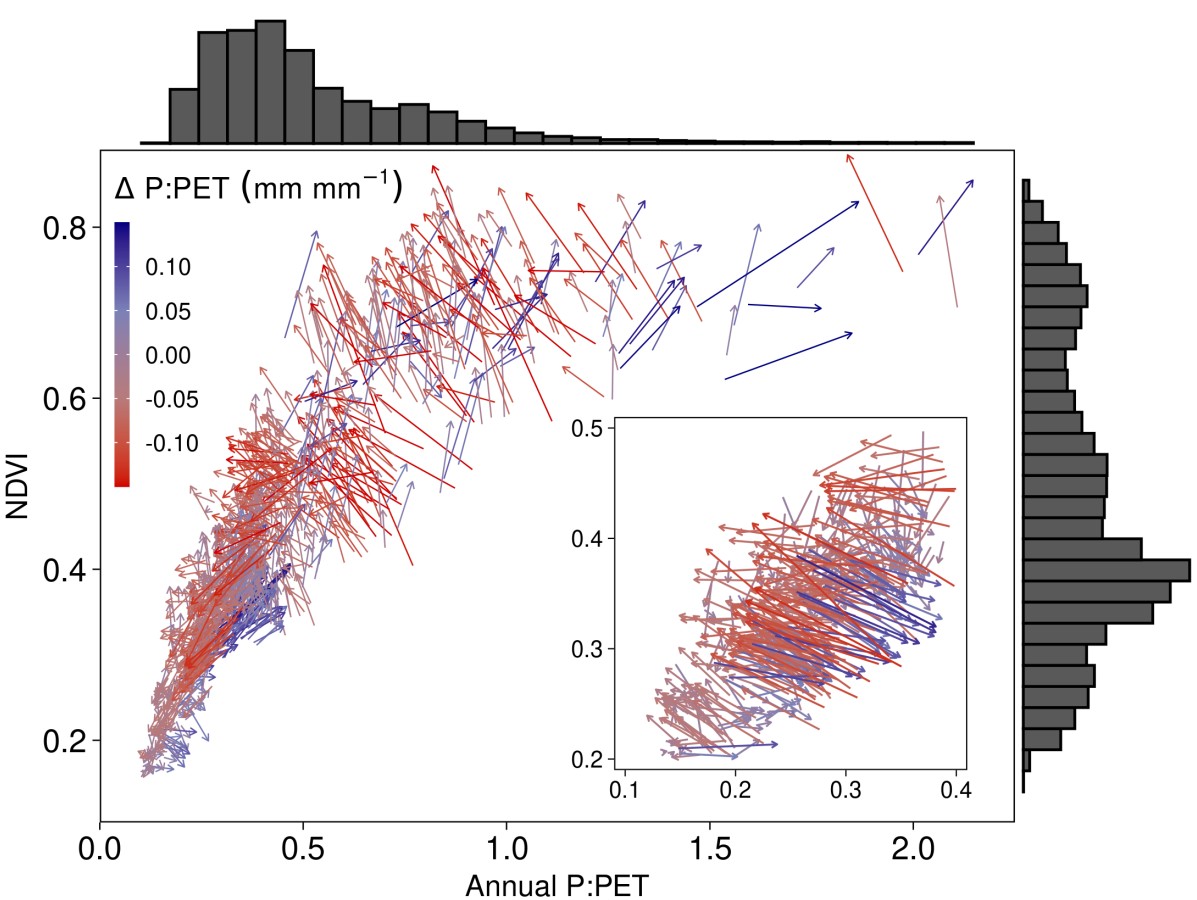

**Figure 1.** Individual grid cell temporal 38-year trajectories of the normalized difference vegetation index (NDVI) and the ratio of annual precipitation (P) to potential evapotranspiration (PET). A vector field plot showing the direction of change in mean annual NDVI and P:PET between 1982-1986 and 2015-2019 for 1000 randomly sampled grid-cell locations (color indicates direction of change in P:PET as indicated by legend). An inset shows a magnification of the samples from the 0.1-0.4 P:PET range. The distributions of mean P:PET $(mm\,mm^{-1})$ and NDVI for the period of 1982-1986 are shown as histograms above and to the right of the main panel. Note that the majority of arrows shift towards higher aridity (lower P:PET) and higher NDVI.

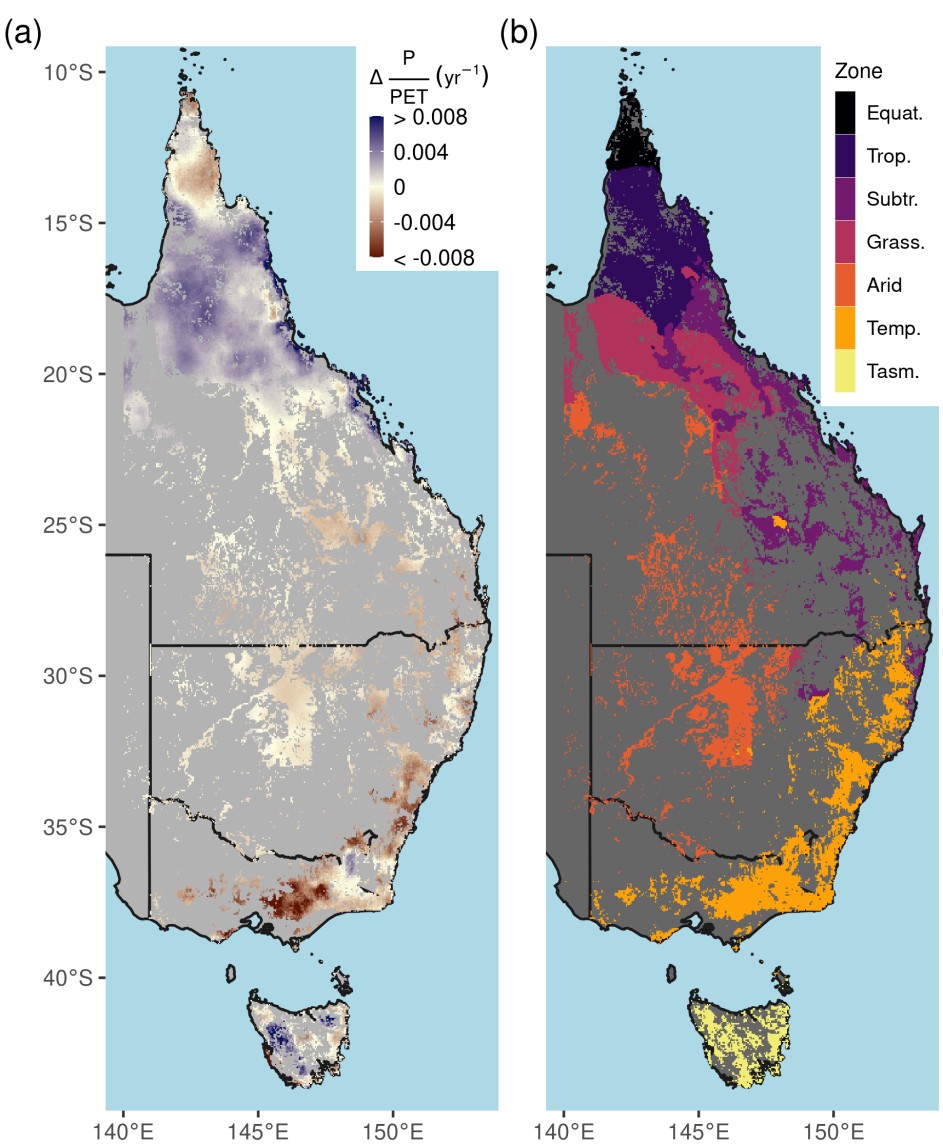

**Figure 2.** Long-term aridity change and climate zones. (a) The linear trend of annual P:PET (the moisture index) between 1982-2019, (b) Simplified Köppen climate zones. Climate zone abbreviations correspond to Equatorial (Equat.), Tropical (Trop.), Subtropical (Subtr.), Grassland (Grass.), Temperate (Temp.), and Temperate Tasmania (Tasm.).





## 3 Results

### 3.1 Long-term Greening in a changing climate

Parts of northern Queensland grew wetter, but aridity (as measured by reduced P:PET; the moisture index, MI) in over 52%
of eastern Australian woody ecosystems since 1982 (Figs. 1,2a). Aridity decreased over northern Queensland encompassing
the entirety of Equatorial and Tropical regions, and most of the Grassland and Subtropical regions, driven by large wet-season
increases in precipitation (Fig. A2a). Widespread increases in PET were evident from September-February (Fig. A2b). At
the same time as these changes in climate were occurring, over 92% of these regions experienced overall greening (Fig.
3a), including regions where P:PET declined (Figs. 1,2b,A3). The relative increases in NDVI were comparable between the
earlier AVHRR epoch (1982-2000) and the later MODIS epoch (2001-2019) at 5.7% (CI=[-2.9%, +20.3%]) and 5.1% (CI=[-
6.4%, +20.1%]), respectively. However, the spatial patterns of greening/browning differed between epochs (Fig. 3b), and most
regions also showed high decadal-scale variability of greening/browning trends (Fig. 4). The overall greening trends between
the AVHRR 1982-2000 epoch and the MODIS 2001-2019 epoch generally agree across regions and seasons. However linear
NDVI trends fit over shorter intervals of 10 years are much less consistent (Fig. 4), exemplifying the importance of estimating
trends over long enough periods to average over decadal-scale variability. Long term browning only occurred in the Arid region
(Fig. 3,4). Nevertheless, by examining NDVI trends over nearly forty years, we were able to separate regional decadal-scale
variability from the overall, broad greening trend across eastern Australia (Figs. 3,4,A3).

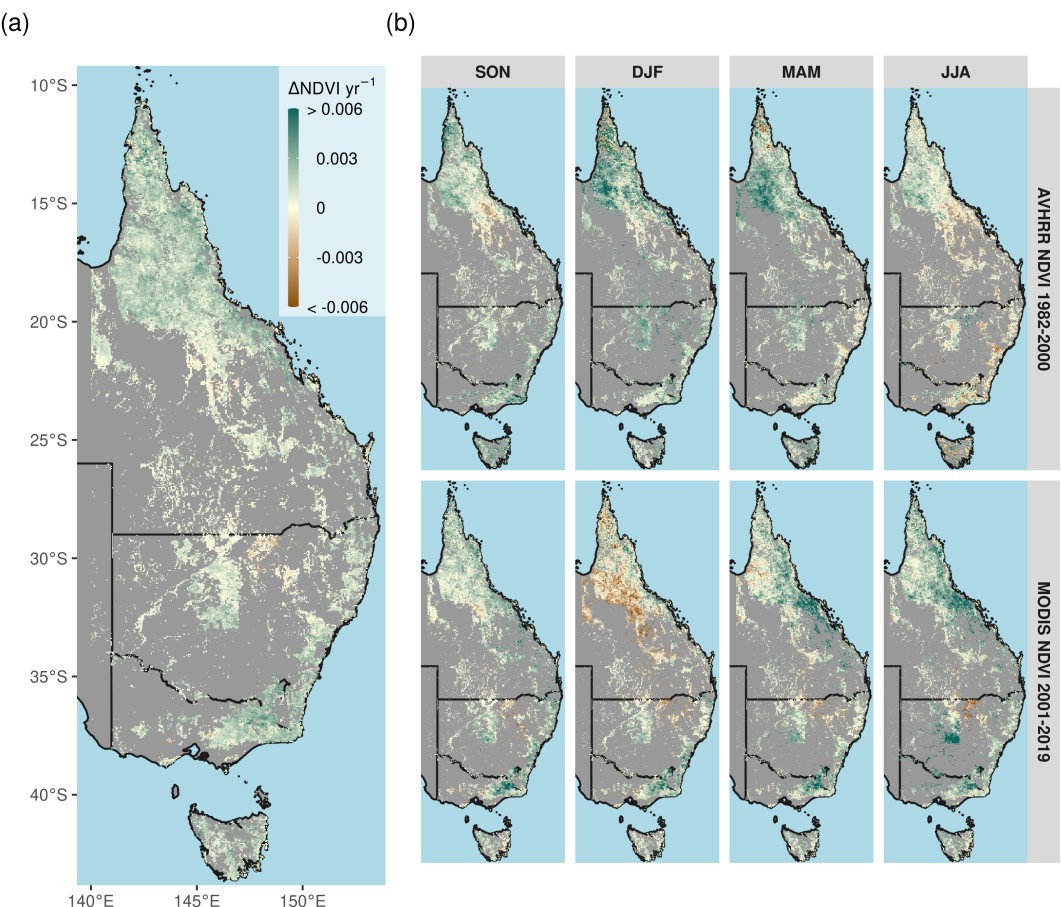

**Figure 3.** Overall long-term NDVI change and change shown by satellite epoch and season. (a) The annual rate of NDVI change from the merged satellite record spanning 1982-2019. The seasonal AVHRR NDVI between 1982-2000 (b-top) and MODIS NDVI between 2001-2019 (b-bottom). Non woody ecosystem regions are masked in gray. A notable browning trend is evident at the interface of the Grassland and Arid regions during DJF of the MODIS time period. Note: Season abbreviations correspond to September-October (SON), December-February (DJF), March-May (MAM), and July-August (JJA).



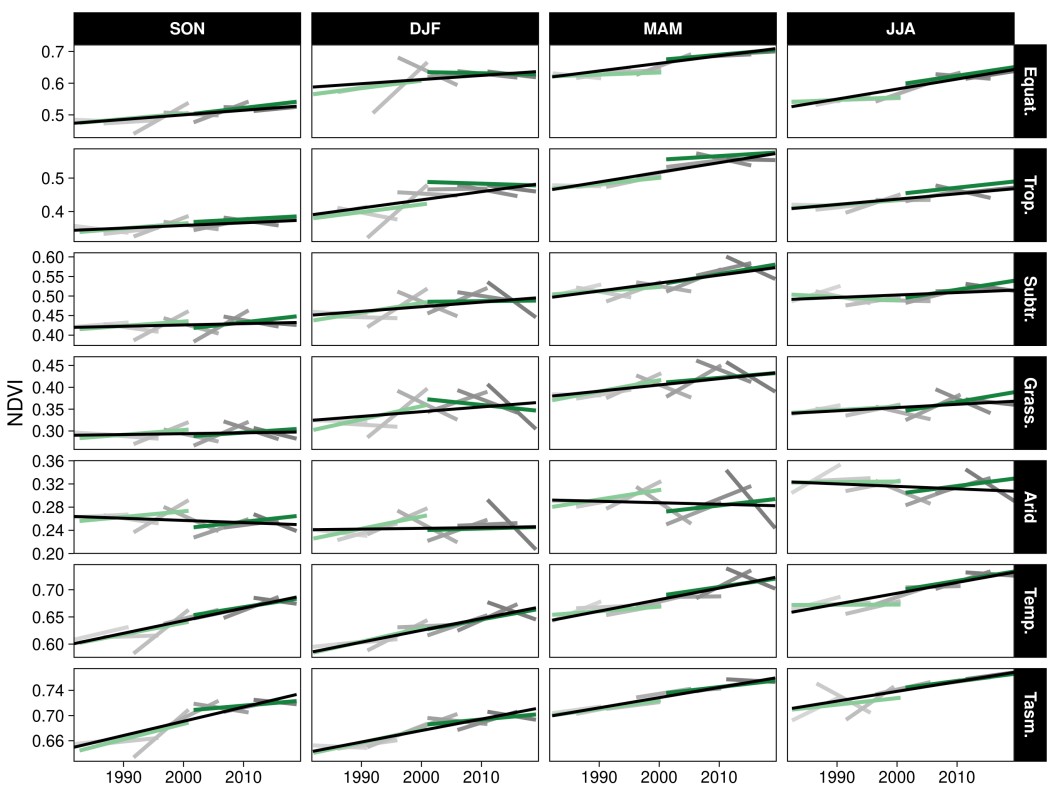

**Figure 4.** Variability of linear trends over varying time periods by season and climate zone. The (black) line represents the overall 1982-2019 trend, the (light green) line represents the calibrated AVHRR 1982-2000, and the (green) line represents the MODIS 2001-2019. Gray colors indicate linear trends from overlapping 10 year time intervals. The boundaries of the climate zones are shown in Figure 7. Note: Climate zone abbreviations correspond to Equatorial (Equat.), Tropical (Trop.), Subtropical (Subtr.), Grassland (Grass.), Temperate (Temp.), and Temperate Tasmania (Tasm.). Season abbreviations correspond to September-October (SON), December-February (DJF), March-May (MAM), and July-August (JJA).





## 3.2 Empirical attribution of the CO$_2$ effect

We found consistently positive NDVI responses to CO$_2$ across the moisture gradient of P:PET for all seasons, with the greatest
increases located in regions of higher P:PET (>0.5) (Fig. 5). The nonlinear Weibull models showed a larger CO$_2$ attributable
effect on NDVI in regions of higher P:PET (Fig. 5), but the effect size of the NDVI response to CO$_2$ was largely consistent
across model forms (Figs. A4-7). The CO$_2$-attributable increase in NDVI between 1982-2019 ranged from approximately 5%
in the Arid interior regions to >20% in the wettest Tropical and Temperate regions (Fig. 5a). This was consistent with linear
model forms when fit for individual grid cell locations (equation 8; Fig. 6), as well as by comparing the CO$_2$ effect size across
16 linear models fit for grid cell locations grouped into bins spanning 0.1 increments of P:PET (equation 7; Fig. A4). The GAM
fit across grid cell locations also indicated a larger CO$_2$ effect in regions with higher P:PET (Figs. 6,7b). Quantile regression
with generalized additive models showed a pronounced response to CO$_2$ across the distribution of pixels with both low and
high NDVI (10 - 97.5 percentiles) across the full aridity gradient of P:PET (Fig. A7).

A recent study found the global CO$_2$ fertilization effect was halved between the 1980s and 2000s (Wang et al., 2020). In con-
trast to the estimates over eastern Australia from wangRecentGlobalDecline2020 over, we found no consistent evidence of a
decline in the effect of CO$_2$ on NDVI through time. Neither the GAM estimates nor the robust linear model estimates of the
CO$_2$ effect showed any consistent evidence of a weakening CO$_2$ effect between 1982-2000 and 2001-2019 (Fig. 6). The central
25-75% percentiles of the distribution of robust linear model effect sizes overlapped in all regions between epochs. The central
25-75% of the GAM estimated distributions also overlapped, with the exception of the Grassland and Arid regions where the
CO$_2$ effect was larger during 2001-2019. Consistent with the finding of a greater CO$_2$ effect in wetter regions (Fig. 5), the
robust linear models and the GAM estimated the CO$_2$ effect to be greatest in the Equatorial and Tropical regions, and lowest
in the Arid region (Fig. 6).



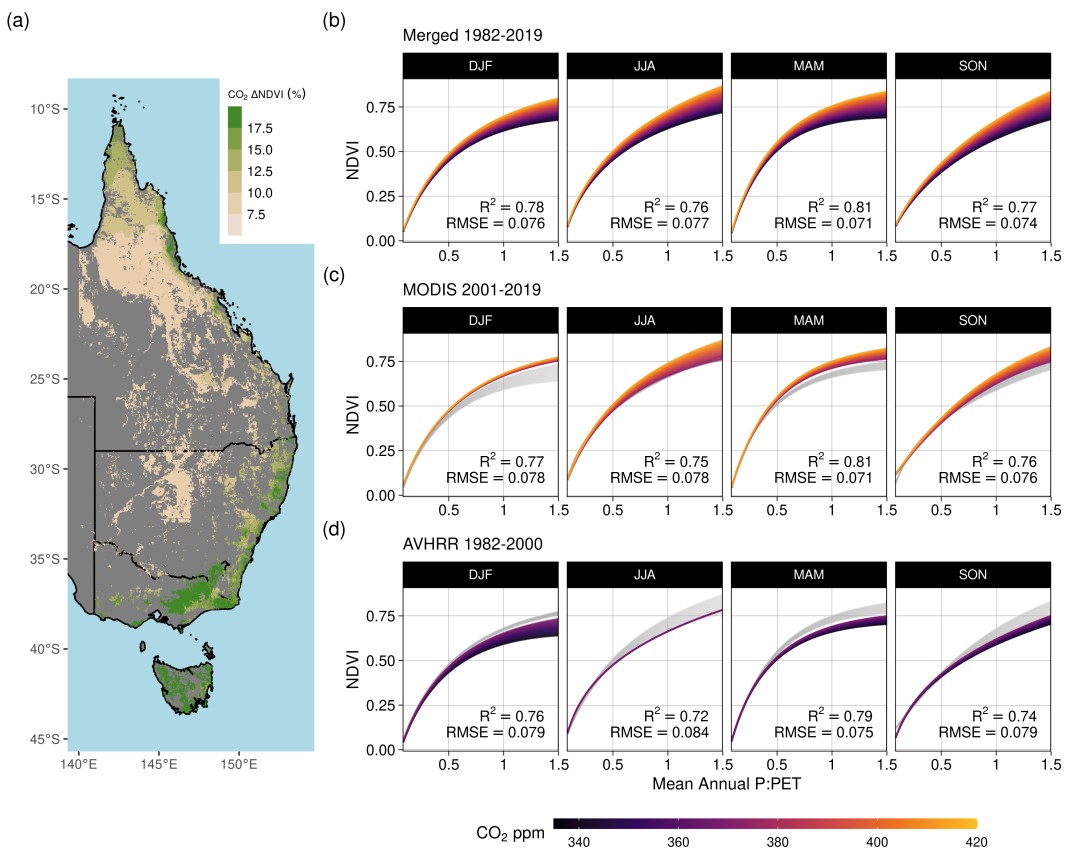

**Figure 5.** Effect of increasing $CO_2$ on seasonal NDVI across P:PET. Predictions of seasonal NDVI as a function of mean annual P:PET fit using a standard Weibull function (methods - eq 2), modified with linear effects of $CO_2$, the running 12-month anomaly of P:PET, and the satellite sensor. The $CO_2$ concentration gradient represents the atmospheric $CO_2$ change between 1982-2019. Panel (a) maps the total predicted contribution of $CO_2$ towards the relative increase of NDVI between 1982-2019 assuming no anomaly of P:PET. Panel (b) shows the merged sensor response between 1982-2019 across the gradient of P:PET, (c) shows the model response when fit using just MODIS MCD43 data between 2001-2019, and (d) shows the response when the model was fit with the recalibrated AVHRR data between 1982-2000. The AVHRR and MODIS satellite epoch NDVI predictions are plotted in gray for panels B and C, respectively.


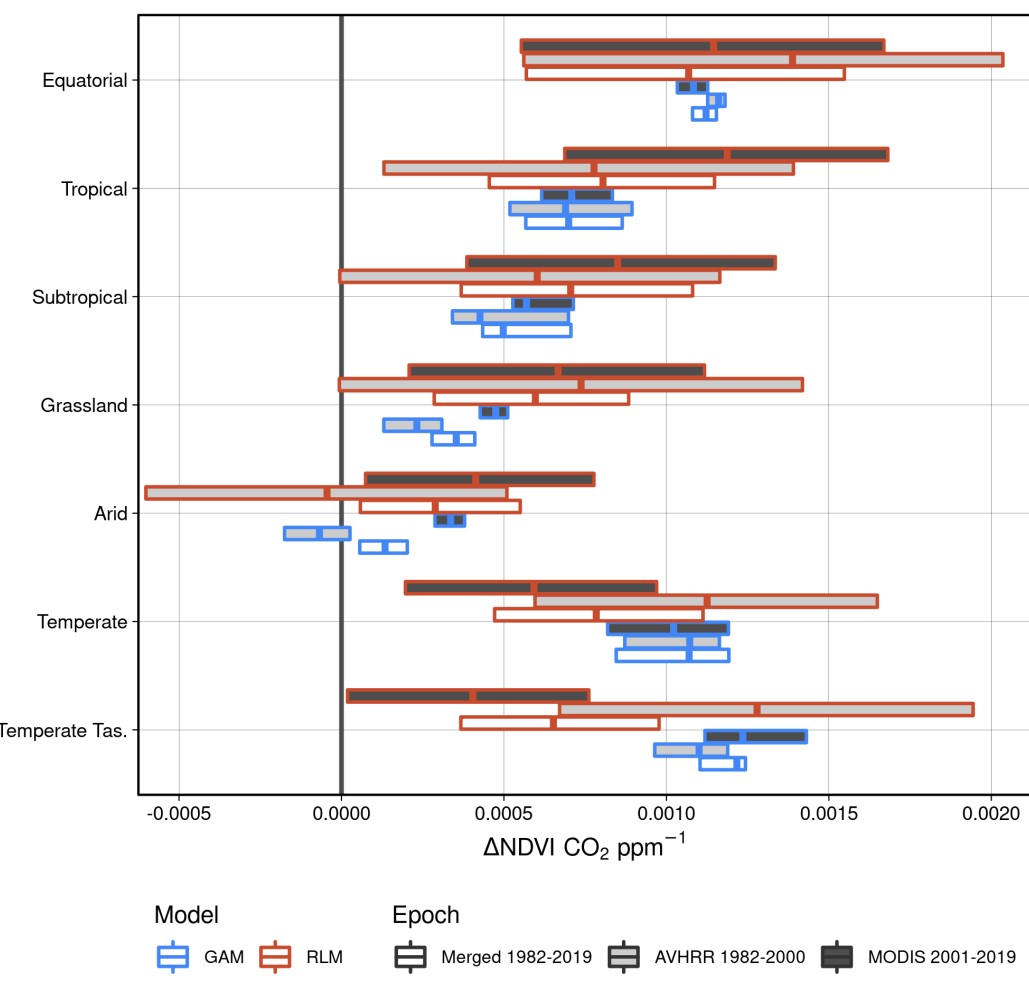

**Figure 6.** Boxplot representation of the $CO_2$ attributable effect upon changing NDVI. Here the 25th, 50th, and 75th percentiles of the $CO_2$ attributable effect on NDVI are shown. The robust linear models (RLM; methods - eq 13) were fit for each individual grid cell location, whereas the generalized additive model (GAM; methods - eq 14) was fit using all grid cell locations. The distribution of RLMs yielded a median $R^2$ of 0.58 and RMSE of 0.025 over the merged period. The GAM had an overall $R^2$ of 0.91 and RMSE of 0.049.





### 3.3 CO₂ driven greening and expectations from water use efficiency

The range of statistically estimated $CO_2$-attributable greening responses was compared with the expectation from the theoretical $CO_2$ water use efficiency model. The Donohue et al. (2013) $CO_2$ x WUE model (see methods) accounts for changes in VPD, but assumes no change in water supply. Assuming an equal split in the benefits of WUE between greater carbon assimilation and increased foliage cover (see below for alternative assumptions), the model predicted an 8.7% (10-90% percentile range [+6.8%,+10.2%]) increase in NDVI (proxy for foliage cover, see methods). This compared to an estimated 11.7%

([+4.6%,+14.6%]) relative increase from the GAM, when accounting for simultaneous increase in VPD (which the WUE accounts for), and factoring out the effects of changing precipitation and PET (which the WUE model does not account for). We needed to assume differing levels of allocation to foliar gain (i.e. not 50%) for the theoretical WUE model to match the statistically estimated $CO_2$ effect (Fig. 7d). Regions of higher P:PET (Equatorial, Tropical, Temperate, and Temperate Tasmanian) required greater allocation fractions than 50%, whereas the allocation fraction would be between 25-50% for regions with

lower P:PET (Arid, Grassland, and Subtropical). In comparison, the PETA hypothesis (Donohue et al., 2017) predicted the greatest $CO_2$ effect on leaf area to be in regions with the lowest LAI, but this was not supported by the statistically estimated $CO_2$ effect on NDVI (Fig. A9). Despite having the lowest LAI, the Arid region received the smallest $CO_2$ effect, but it is worth noting the Arid region also experienced the greatest increase in VPD and reduction in P over the 38 year period (Fig. 7a, A8). In contrast, the largest estimated $CO_2$-attributable effect on greening were found to be in the Equatorial and Tropical areas.

### 285 3.4 Co-occurring shifts in aridity, NDVI, and vegetation cover

The shifts in P:PET and NDVI were accompanied by vegetation cover changes in some regions. Most notably, the Arid and Temperate regions experienced the strongest zonally averaged declines in P:PET (Fig. 8a) and increases in VPD (Fig. A8). Seasonal greening trends were relatively similar apart from the aforementioned exceptions in the Grassland. The MODIS vegetation continuous fraction data from 2001-2018 indicated most regions experienced modest changes in tree vegetation cover.

The largest decline in tree cover occurred in the Temperate regions (Fig. 8c,A10. Most regions experienced declines in non-vegetated (bare) cover, increases in non-tree vegetation, and modest change in tree-cover (Fig. 8c), however the proportional increase of non-tree vegetation typically exceeded tree cover increases (Fig. A10).





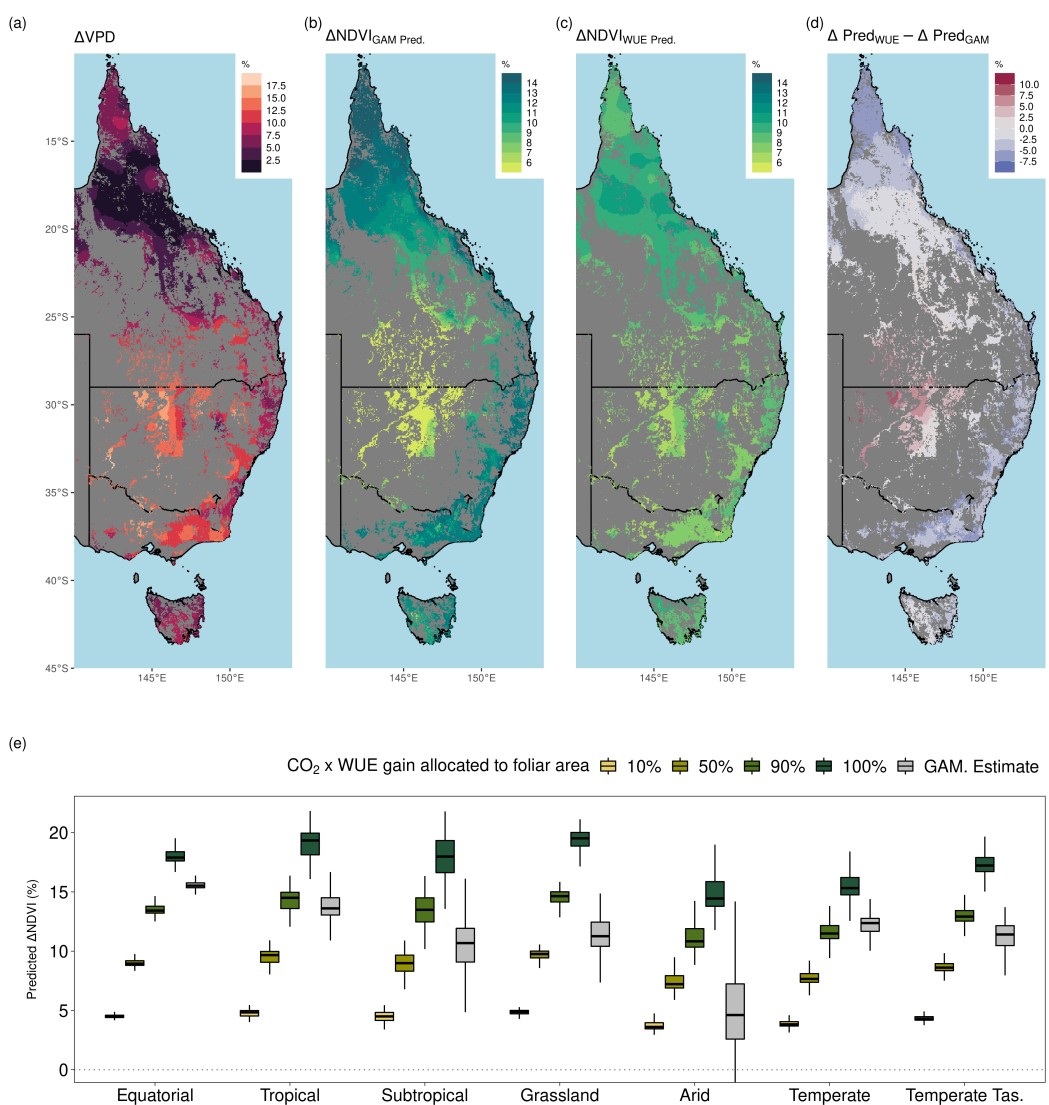

**Figure 7.** Long-term changes in vapor pressure deficit, and comparison between the $\Delta NDVI$ due to $CO_2$ and the expected $CO_2$ fertilization effect on foliar area due to gains in water use efficiency (WUE). (a) The relative increase in the annual mean of vapor pressure deficit (VPD) between 1982-2019. (b) The predicted relative increase in NDVI due to $CO_2$ (methods - eq 14) with a concurrent increase in VPD. (c) The relative expected increase in NDVI following the theoretical WUE prediction where 50% of the gain is allocated to foliar area (methods - eq 11).

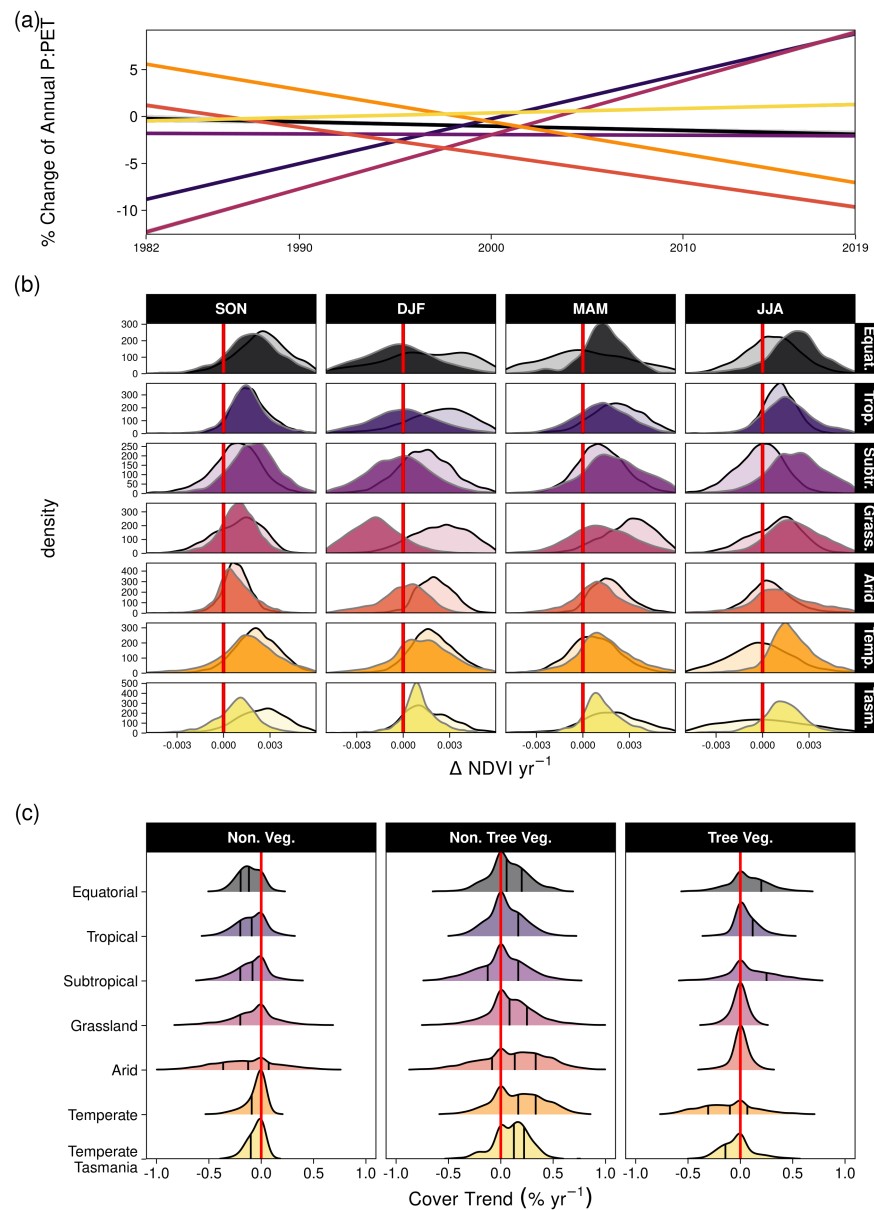

**Figure 8.** The relative percentage change in P:PET by climate zone and corresponding distributions of $\Delta\ NDVI\ yr^{-1}$ and percentage annual change in vegetation cover fraction. (a) The linear relative changes in annual P:PET trend by climate zone between 1982-2019, as estimated by robust regression (methods). (b) Distribution of linear long term NDVI trends for the six climate clusters by season using the Theil-Sen estimator. Filled distributions are trends from the MODIS sensors (2001-2019) and transparent (black outline) distributions are from the AVHRR sensors (1982-2000). (c) Distributions of the linear pixel level trends using the Theil-Sen estimator for non-vegetated cover, non-tree vegetation cover, and tree cover between 2000-2018. The 25, 50, and 75% quantiles are overlaid.

Note: Climate zone abbreviations are as follows: Equatorial (Equat.), Tropical (Trop.), Subtropical (Subtr.), Grassland (Grass.), Temperate (Temp.), and Temperate Tasmania (Tasm.).



## 4 Discussion

### 4.1 Australian woody vegetation as model systems to quantify $CO_2$ fertilization

Australia is ideal to explore the $CO_2$ contribution towards vegetation greening because there are fewer confounding effects to drive greening. Forest trees are evergreen and the growing season is rarely limited by temperature or radiation. The study region spans a large moisture gradient (Fig. A1a), but unlike much of the global tropics, the large majority of the study region is not so cloudy as to prevent multiple high quality multispectral satellite retrievals per month. Australia has also not been subjected to other prominent drivers of greening such as nitrogen deposition (Ackerman et al., 2019). Nevertheless, Australia

has experienced notable land-use change during the study period such as high rates of deforestation in Queensland and northern New South Wales (Evans, 2016). However, we excluded affected pixel locations from the analysis.

Prior global analyses on warm arid environments have quantified the $CO_2$ effect on greening (Donohue et al., 2013), yet a more expansive global-scale analysis of all terrestrial vegetation using dynamic global vegetation model attribution did not connect Australia's greening to changes in atmospheric $CO_2$ concentration (Zhu et al., 2016). Australian studies documented

the greening trend up to 2010 using the long-term AVHRR record (Donohue et al., 2009; Ukkola et al., 2016) and have been able to partially attribute $CO_2$ as a driver of greening in sub-humid and semi-arid regions. Here we advanced upon prior research to separate the effects of disturbance, and changes in aridity and moisture in order to quantify the $CO_2$ fertilization effect across the full spectrum of moisture availability experienced by Australian woody ecosystems, notably for 38 years.

### 4.2 Regional differences in greening and browning through time

Despite the region's high decadal scale variability of NDVI (Fig. 4), the nearly four decade long record allowed us to separate the $CO_2$ effect on NDVI from the anomalies caused by drought (e.g. 2003-2009) or high rainfall (e.g. La Niña 2010-2011). Although the long-term greening trends we document in the nine years following earlier studies are generally consistent (Donohue et al., 2009; Ukkola et al., 2016), our results diverge and lead to key differences in interpretation of why NDVI has continued to increase. First, while the relative increase in NDVI between the 1982-2000 AVHRR epoch (5.7%) and the 2001-

2019 MODIS epoch (5.1%) are comparable (Figs. 3b,8b), the underlying reasons for the change differ. VPD changed minimally between 1982-2000, whereas it rapidly increased between 2001-2019 (Fig. A8). These increases were largest in the most Arid and Temperate regions (12.7% and 11% since 1982, respectively; Fig. 6a), and when coupled with seasonal reductions in precipitation (Fig. A2) these would have partially offset benefits from increased intrinsic water-use efficiency (equation 17). In contrast, more than half of the Equatorial, Tropical, Subtropical and Grassland regions in northern Queensland experienced

increases in precipitation since 1982 (Fig. A2; Ukkola et al. (2019)), thus allowing these locations to exceed the predicted NDVI increases from the WUE model (Fig. 7d).

It should be noted not all regions experienced consistent greening trends throughout the observation period. For example, 'greening' shifted to 'browning' during the austral summer (Dec - Feb) in the Arid and Grassland regions of Queensland between the 1982-2000 and 2001-2019 records (Fig. 3b). It is unclear why browning occurred during austral summer time

in the Grassland region (Figs. 3b,4,8b). The declines in NDVI during 2001-2019 may have been meteorologically driven and





related to shifts in the distribution of wet and dry season precipitation (Fig. A2). Alternatively, the shift could be due to changes in fire and cattle management that have been particularly prevalent across regions of Queensland in recent decades (Seabrook et al.). Further, greening may have been suppressed in parts of Queensland because cattle ranching activity has intensified and has driven forest conversion to managed pasture in the region (McAlpine et al., 2009).

## 4.3 Attributing a $CO_2$ fertilization contribution towards greening

Plants increase their rates of photosynthesis in response to rising atmospheric $CO_2$, whilst also reducing stomatal conductance, which reduces evaporative losses and combined, the two responses lead to greater WUE (Ainsworth and Rogers, 2007; Morison, 1985). In water-limited ecosystems, it has been hypothesized that this physiological response by plants to $CO_2$ should result in increased leaf biomass (Donohue et al., 2013; Ukkola et al., 2016). While all our statistical approaches indicated a year round positive $CO_2$ effect, in contrast to theory the effect was consistently greater in regions with higher P:PET (Figs. 5-7;A4-7). Our analysis also diverges with a global coarse-scale analysis that found weakening of the $CO_2$ fertilization effect across both southeastern and northern Australia (Wang et al., 2020). We found no meaningful difference in the $CO_2$ attributable effect towards greening between the AVHRR 1982-2000 and MODIS 2001-2019 epochs to support the finding of temporally weakening $CO_2$ effect (Fig. 6). The WUE model predicted similar relative rates of NDVI increase between the two epochs (4.5% and 4.7% for AVHRR and MODIS), yet for different reasons. $CO_2$ increased by 30 ppm between 1982-2000 while VPD changed minimally and precipitation increased in Queensland and the Arid region (Fig. A8); all of which are favorable to increasing NDVI. In contrast, the larger $CO_2$ increase between 2001-2019 (40 ppm) was offset by a spatially ubiquitous increase of VPD (3.7%, CI=[-0.4%,8.6%]; Fig. A8) and the occurrence of two multi-year droughts (Millennium Drought 2003-2009, and The Big Dry 2017-2020). Despite the widespread evidence of the $CO_2$ effect, the WUE model notably underpredicted greening in some regions (Fig. 7).

## 4.4 Deviations from WUE

We found the $CO_2$ effect on foliar area was the smallest in the driest climate regions (Figs. 5-6,A4-7). This was at odds with the WUE prediction (at 50% allocation; Fig. 7e), and contrary to expectation that the greatest WUE-derived benefit from $CO_2$ would be in drier climates (Donohue et al., 2017; McMurtrie et al., 2008). These deviations may have resulted from ecosystems processes beyond the scope of a simple model, such as the phenology of the vegetation composition, and disturbances not captured by the satellite products (e.g. small fires, grazing). Browning in the Grassland region (Fig. 3b,A3) may have been caused by distinct dry season phenological differences between overstory woody vegetation and understory $C_4$ grasses (Moore et al., 2016), which are dominant there (Murphy and Bowman, 2007). $C_4$ grasses may have been favored over $C_3$ because precipitation increased during the austral summer over the course of the study (Fig. A2) and this higher concentration of rainfall during the warmest months is thought to favor $C_4$ grasses (Hattersley, 1983; Knapp et al., 2020; Murphy and Bowman, 2007). Finally, the linear dependency of leaf area upon VPD in the theoretical model may be ill-suited for extreme anomalously arid conditions because NDVI observations suggest a strongly nonlinear relationship with large VPD anomalies (Fig. A11).





We explored how much of the higher WUE benefit would have to be allocated to match the GAM estimated $CO_2$-attributed changes in NDVI. Allocation rates far greater than 50% would be required to match the WUE prediction in the Tropical and

Equatorial zones (Fig. 7e), whereas allocation would need to be less than 50% to match the GAM estimate in the Arid region. The Arid region experienced the greatest relative increase in VPD (Fig. 7a, A8), yet the theoretical model still predicted a small but positive increase in NDVI (Fig. 7c) which the GAM estimate suggests would be closer to a 10% rather than 50% foliar allocation level from the WUE benefit (Fig. 7d). Nevertheless, the smaller effect over the Arid region was consistent with earlier observational findings across Australia (Ukkola et al., 2016) and experimentation from the Nevada Desert FACE

experiment (Smith et al., 2014).

### 4.5   Relation to ecosystem $CO_2$ fertilization experiments

Notably, a four year long ecosystem-scale $CO_2$ manipulation experiment carried out in a mature Eucalyptus woodland in Sydney (EucFACE) did not observe an increase in leaf area under elevated $CO_2$ (Jiang et al., 2020b). The experimental site is located upon phosphorus poor soils, typical of Australia. The lack of a leaf area growth response observed at EucFACE is not

necessarily inconsistent with the greening effect observed in this study. Our observational window of 38 years is much longer than the elevated $CO_2$ exposure time in the experiment (four years in Jiang et al., 2020b), covers different $CO_2$ increments (historical vs future) and could imply that woodlands are eventually able to liberate belowground phosphorus to support greater biomass growth on longer timescales. Increased autotrophic soil respiration and belowground productivity were observed at EucFACE under elevated $CO_2$ exposure (Drake et al., 2016; Jiang et al., 2020b), as was a brief period of enhanced nitrogen

and phosphorus mineralization (Hasegawa et al., 2016). Over time, this increased investment of carbon belowground could potentially liberate sufficient phosphorus to support an expansion of leaf area. The reduced allocation to foliar area in the Arid region (Fig. 7e) may reflect that extra carbon derived from $CO_2$ is allocated belowground to increase water uptake or mitigate other resource limitations such as soil phosphorus (Jiang et al., 2020a).

### 4.6   Vegetation composition shifts

Most grid-cell locations in the Temperate zone experienced simultaneous apparent declines in tree cover and increases in non-tree vegetation cover (e.g. grasses/shrubs) (Figs. 8c, A10). This is surprising because we focused this regression analysis on 2001-2018 in order to exclude the reduced tree cover due to the catastrophic megafires of 2019/20 (Nolan et al., 2020). Some may question the veracity of the MODIS Vegetation Continuous Fraction product (DiMiceli et al., 2017) to accurately distinguish Australian tree cover from non-tree vegetation, however this pattern is consistent with a recent LiDAR derived tree-

cover time series of Australia (Liao et al., 2020). This suggests the drought starting in 2017 was already killing trees prior to the 2019/20 megafires. Field observations of tree decline remain relatively rare, but a citizen science initiative has documented more than 300 locations of non-fire related mass tree mortality between 2018-2020 (of Living Australia", 2020). Similarly, a study using an experimental constrained plant hydraulics model to predict the regions at risk of drought-induced tree mortality, found greater risk in the same arid regions of southeast Australian forests and woodlands (De Kauwe et al., 2020). These

predicted regions of mortality coincide with where we document the greening trends that fell short of the theoretical WUE



expectation. These rapid shifts in vegetation underlie the need for greater continuous field vegetation monitoring to capture change imposed by climate extremes.

## 5 Conclusions

We separated the effects of disturbance and meteorological anomalies with statistical models to show increasing $CO_2$ produced
nearly four decades of widespread vegetation greening across eastern Australia. The large agreement between a theoretical model and the statistically estimated $CO_2$ effect indicated that greening resulted through an increase in water use efficiency. Vegetation greening occurred despite a highly variable and increasingly arid climate, and on soils particularly poor in phosphorus which have likely acted as a constraint on growth. While rising atmospheric $CO_2$ ameliorated what would have been a browning woody ecosystem response to declining P:PET, the $CO_2$ effect was insufficient to promote greening when both P
and P:PET experienced long-term decline, as observed in the more arid regions in our study. Further, it is unknown whether further increases of atmospheric $CO_2$ will continue to enable vegetation to mitigate increases of aridity and VPD under future warming. It is also unclear if trees or grasses are the primary contributors to the recent greening trend. Future localised work is urgently needed to better understand recent changes in tree and grass competition under an increasingly arid climate, which will be essential to help forecast ecosystem resilience. Finally, our results have important implications for understanding Aus-
tralia's terrestrial water availability. Greening trends signal changes in evapotranspiration and runoff, and therefore need to be considered in planning for future land and water resource management on the world's driest inhabited continent.

*Code and data availability.* All data used are publicly available from sources listed in Table A1. Processed data used in model fitting and figures can be accessed via Zenodo data repository: [future repository link]. A git repository with all associated data processing, analysis, and code to reproduce figures is located at: https://github.com/sw-rifai/eastern-Australia-CO2-NDVI-change.

**Appendix A: Figures and tables in appendices**



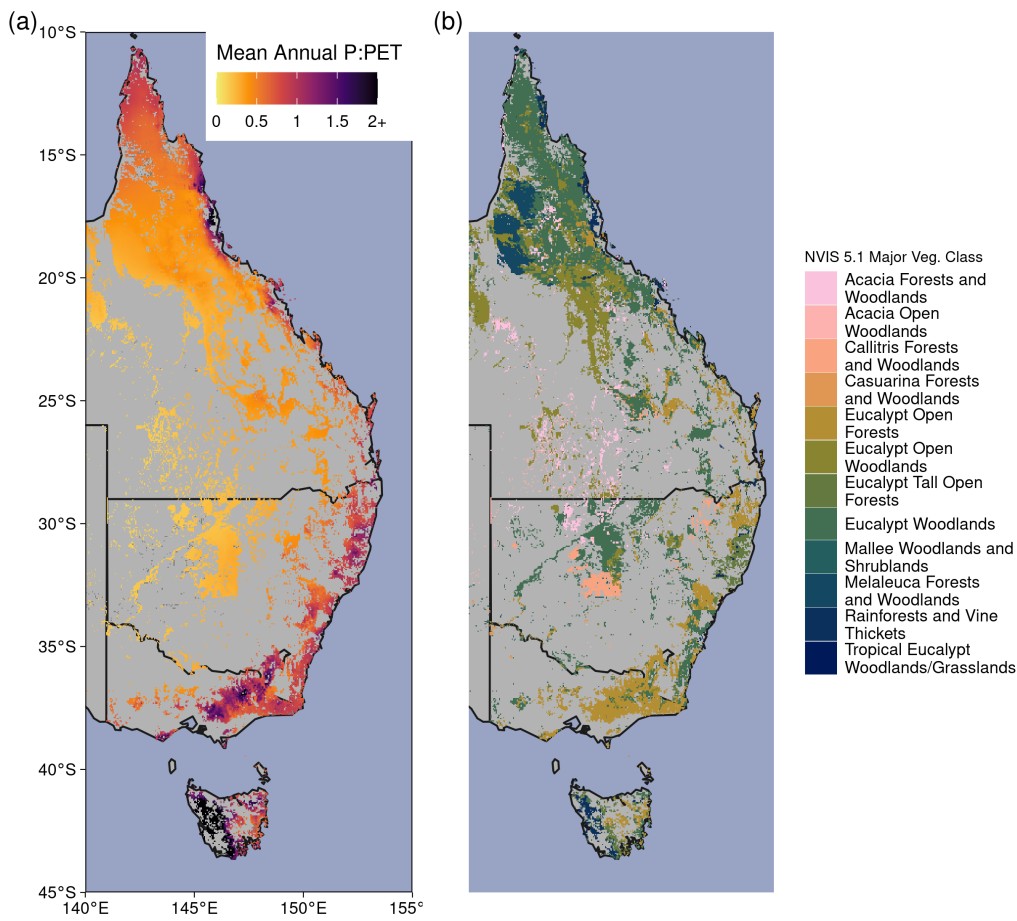

**Figure A1.** (a) Mean Annual P:PET between 1982-2019. (b) Forest and woodland vegetation classes from version 5.1 of the National Vegetation Information System (source: https://www.environment.gov.au/land/native-vegetation/national-vegetation-information-system).


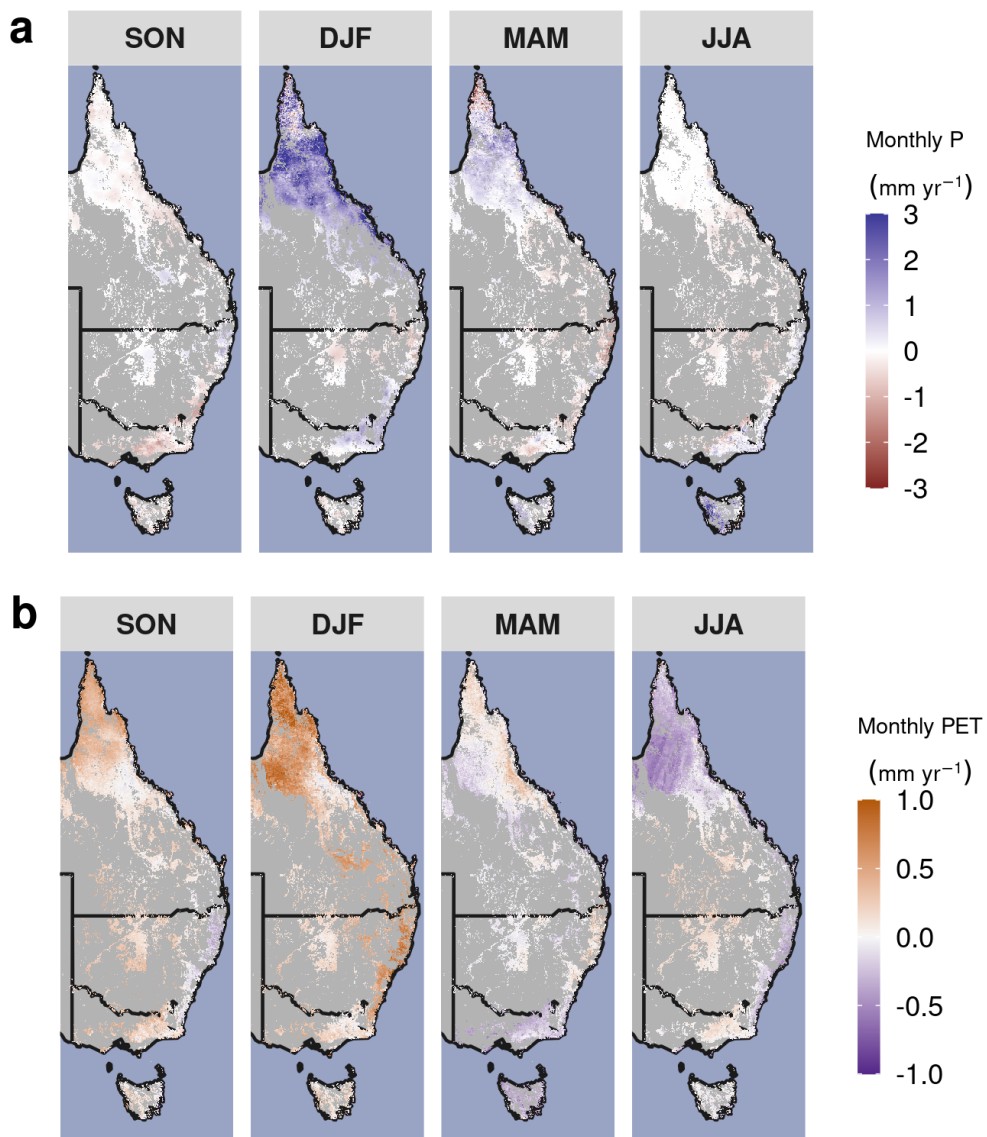

**Figure A2.** Long-term seasonal changes in precipitation (P) and potential evapotranspiration (PET). Linear trend in (a) monthly precipitation and (b) PET by season over the period 1982-2019. Non forest and woodland regions are masked in grey.





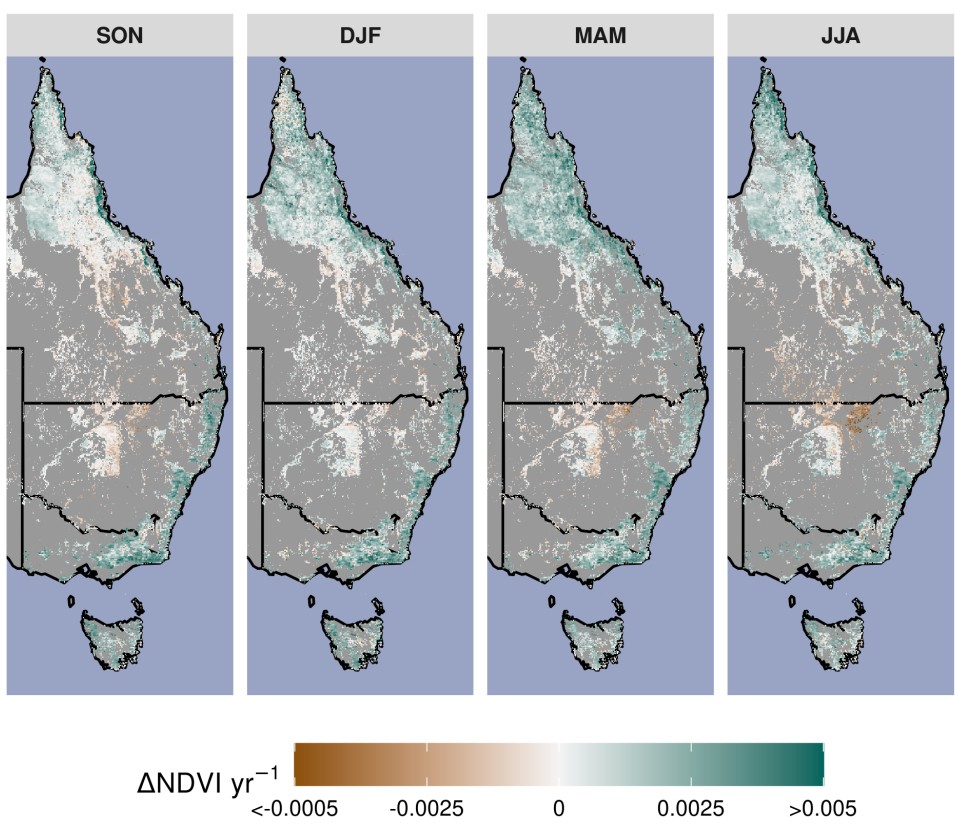

**Figure A3.** The long term seasonal NDVI linear trend between 1982-2019. The Theil-Sen robust linear trend estimator is used with the calibrated merge of the CDR AVHRR (1982-2000) and MODIS MCD43 (2001-2019) surface reflectance products.


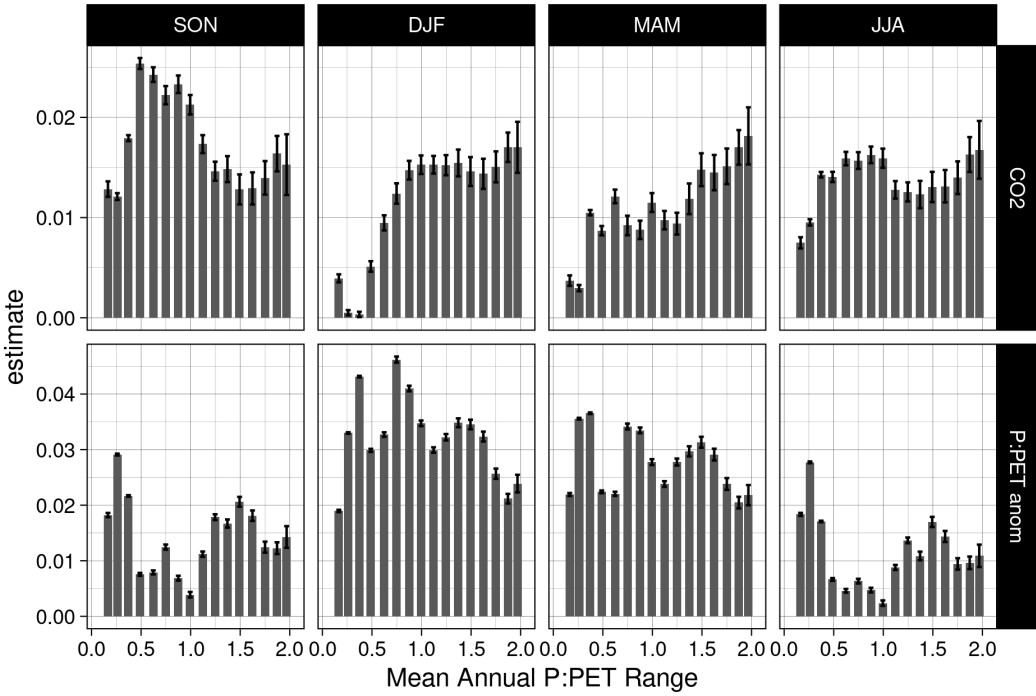

**Figure A4.** Linear model covariate estimates for 128 linear models. The mean annual P:PET range is discretized over 16 increments where NDVI is modeled as a linear function of CO2, $P : PET_{anom.}$, the NVIS vegetation class, and the sensor. Error bars indicate 2x the standard error of the estimate.



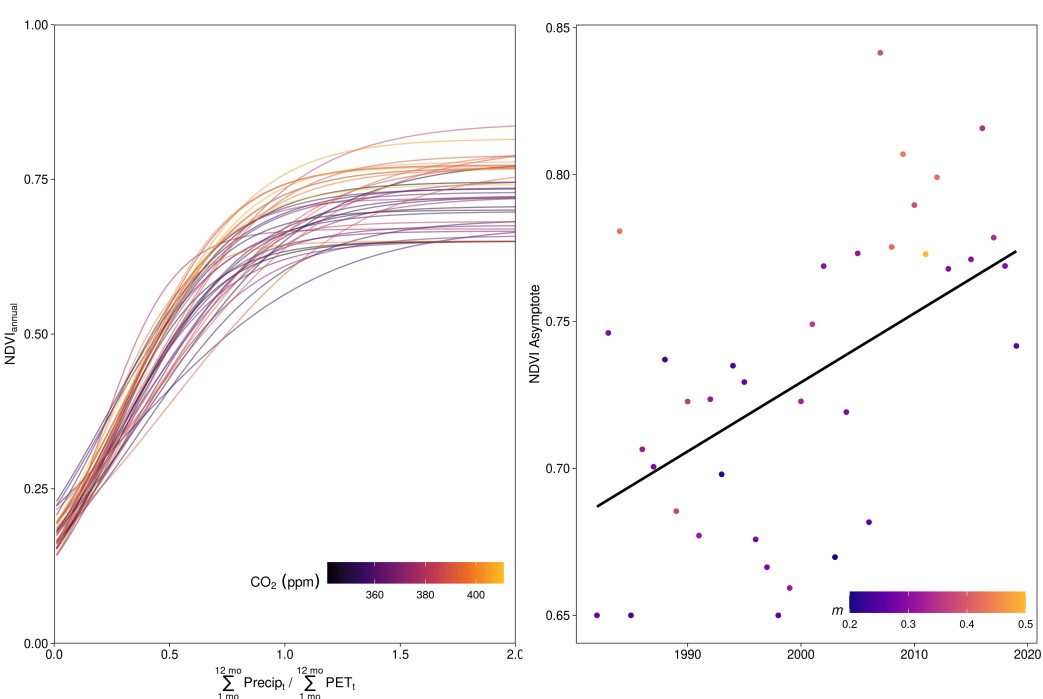

**Figure A5.** NDVI as a logistic function of annual P:PET. (left) Logistic function models, fit per each hydrological year. (right) The positive long-term trend in the NDVI asymptote from the logistic model fits. $m$ is the value of P:PET at the inflection point, and NDVI Asymptote corresponds to the $V_A$ parameter in equation 5.





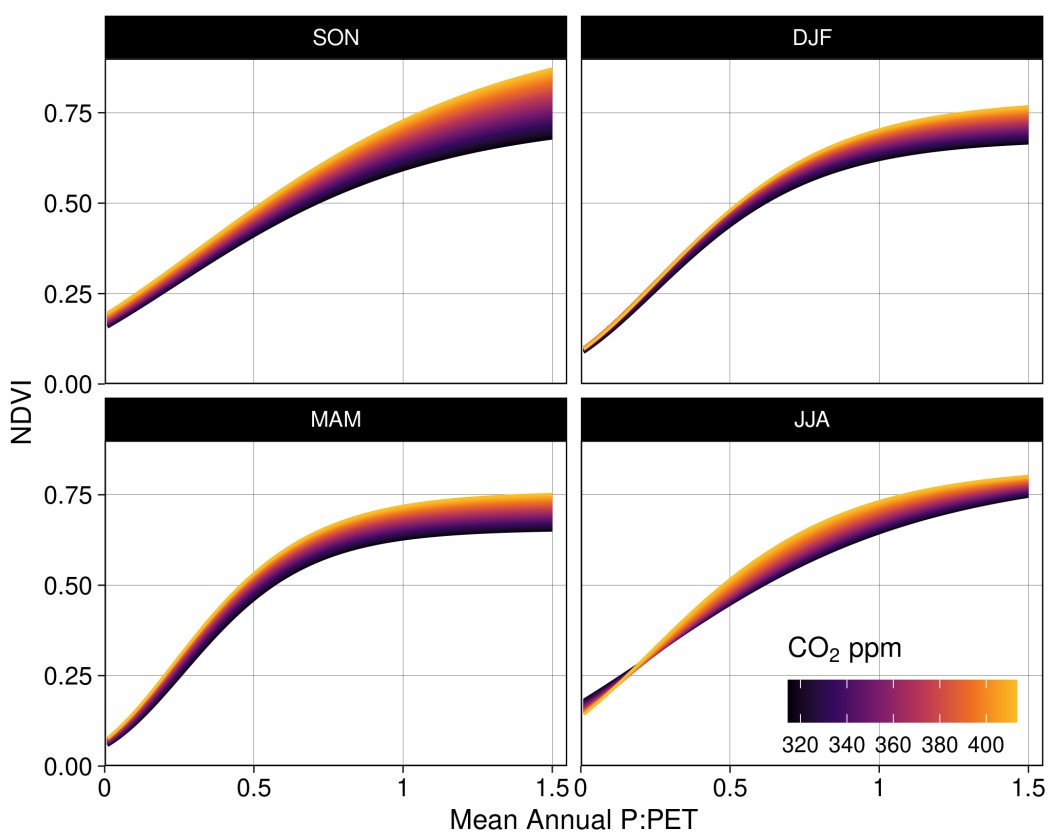

**Figure A6.** Seasonal NDVI as a Richards growth function with linear modifiers of CO2 and the anomaly of P:PET (equation 6).



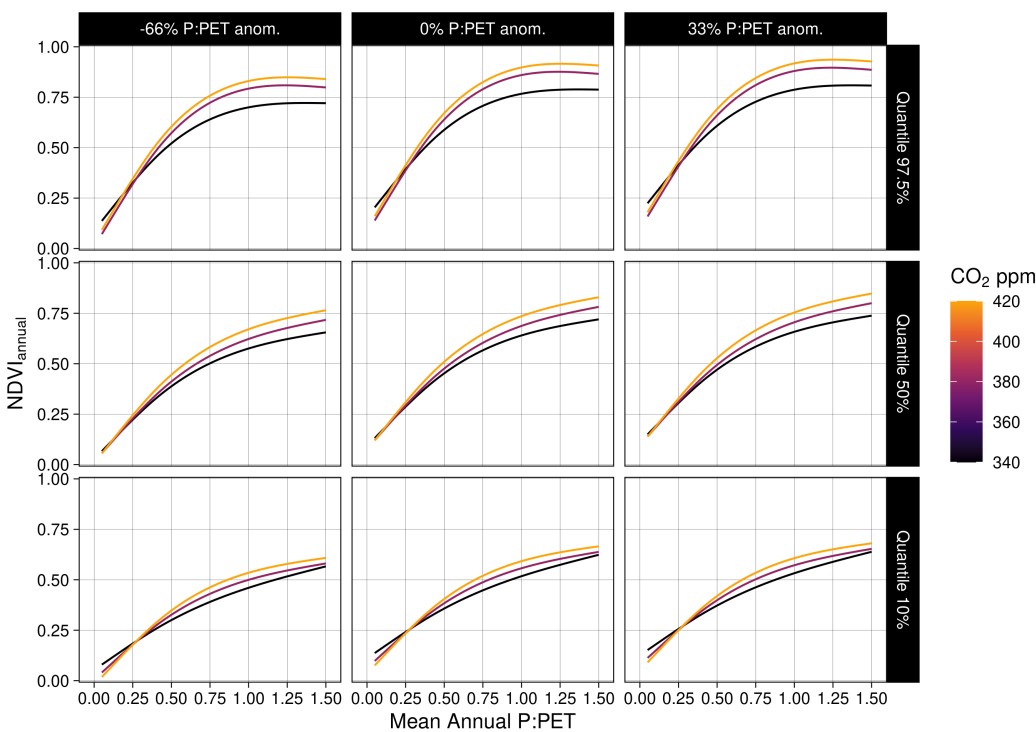

**Figure A7.** Quantile GAM regression predictions across CO2 and anomalies of annual P:PET.

$$NDVI_{3mo} = s(CO_2) + s(\frac{P}{PET_{MA}}) + s(CO2, \frac{P}{PET_{MA}}) + s(\frac{P}{PET_{anom.}})$$

Here 's' represents a thinplate spline smoothing function, $\frac{P}{PET_{MA}}$ is the 30-year mean annual P:PET, and $\frac{P}{PET_{anom}}$ is the annual anomaly of $\frac{P}{PET}$.





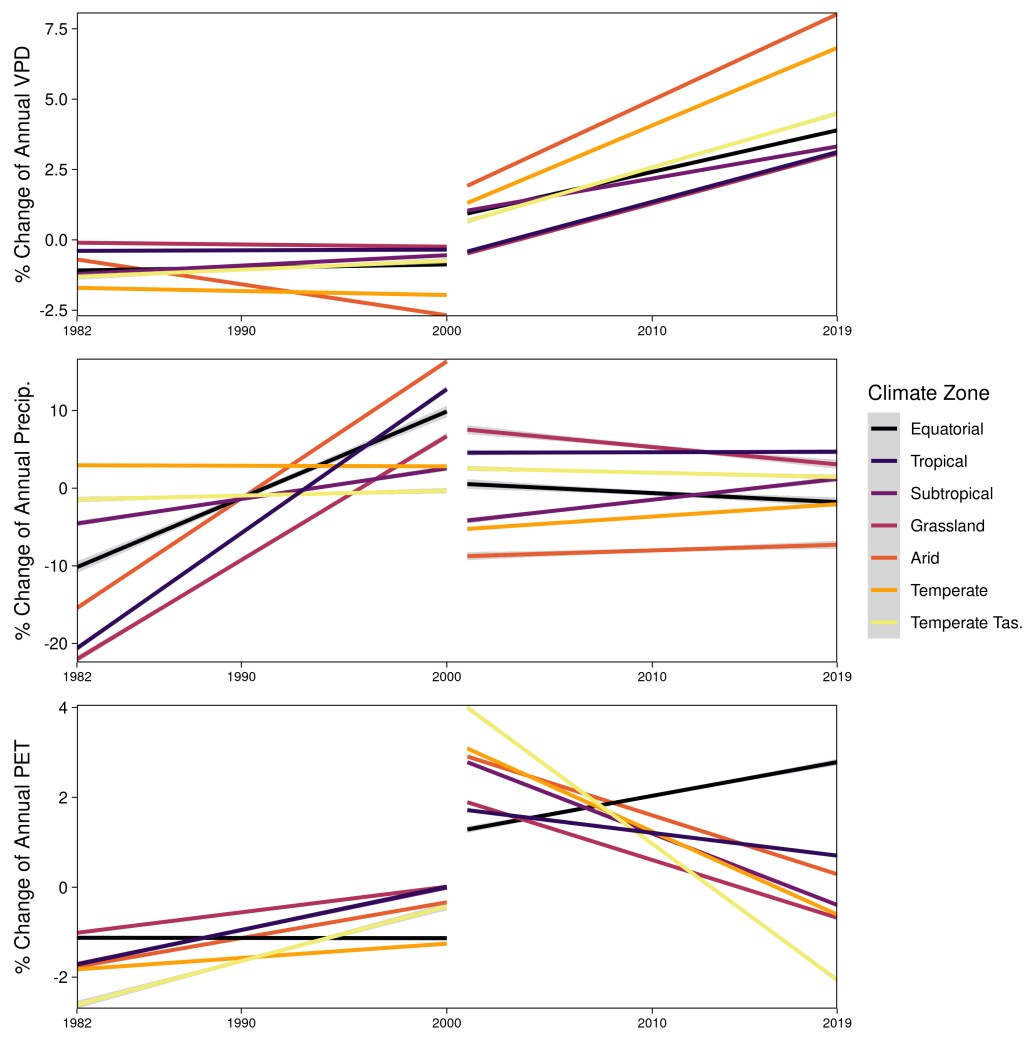

**Figure A8.** The linear relative changes in annual vapor pressure deficit (VPD), precipitation (Precip.), and potential evapotranspiration (PET) by climate zone between satellite epochs AVHRR 1982-2000 and MODIS 2001-2019, as estimated by robust regression. The relative changes are with respect to a climatology period calculated between 1982-2011.

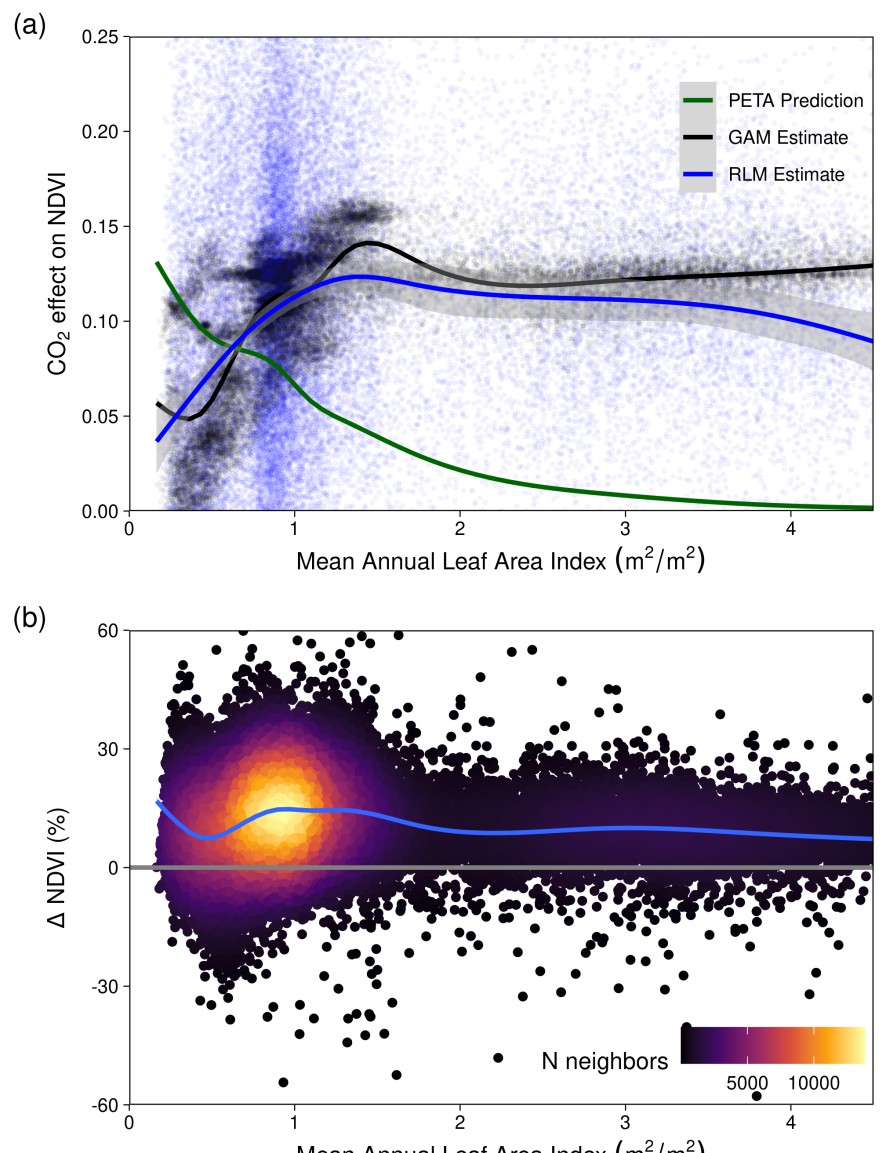

**Figure A9.** (a) The fractional increase of NDVI from CO2 is plotted following predictions from the Partitioning of Equilibrium Transpiration and Assimilation (PETA) hypothesis is plotted (green), the Generalized Additive Model (GAM; methods eq. 6) estimates, and the pixel-level Robust Linear Model (RLM; methods eq. 5) estimates. Note we substitute NDVI for the leaf area formulation of the PETA hypothesis (Donohue et al., 2017) because we assume a linear relationship between relative NDVI increase and relative leaf area increase (Donohue et al., 2013; Carlson & Ripley 1997). (b) The relative change in NDVI between 1982-2019 across the range of mean annual leaf area index calculated with MCD15A3H from 2002-2019.

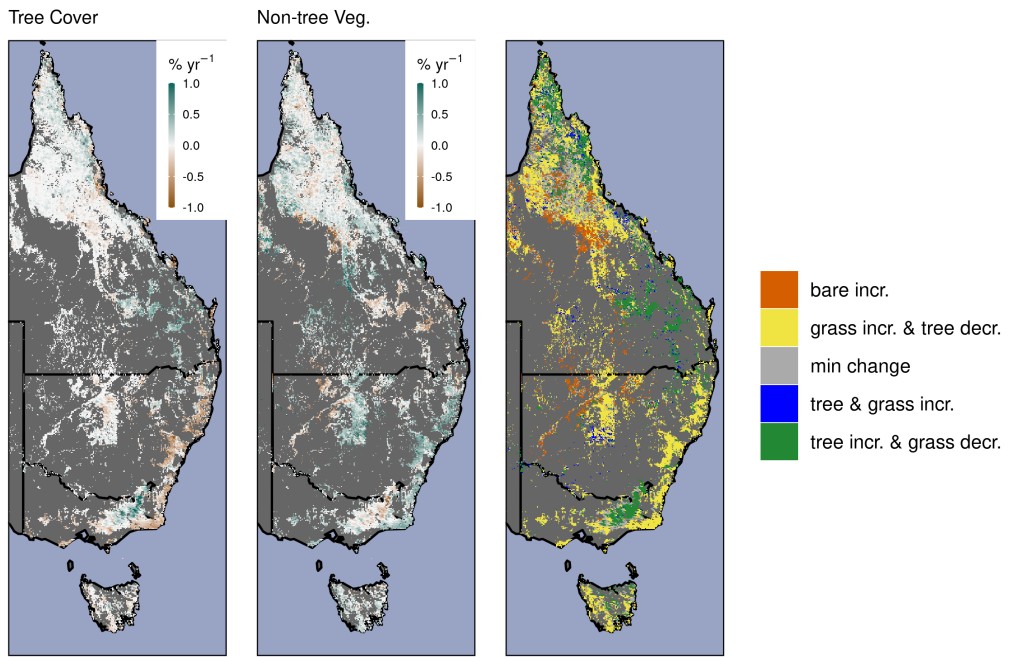

**Figure A10.** The relative shifts of land fraction by tree cover, non-tree vegetation cover (predominantly grasses), and bare ground. 2000-2018 annual rate of change for tree cover (left), non-tree vegetation (middle), and discretized change classes. Min. change indicates the annual rate of change of percentage cover was between -0.1 & 0.1% $yr^{-1}$



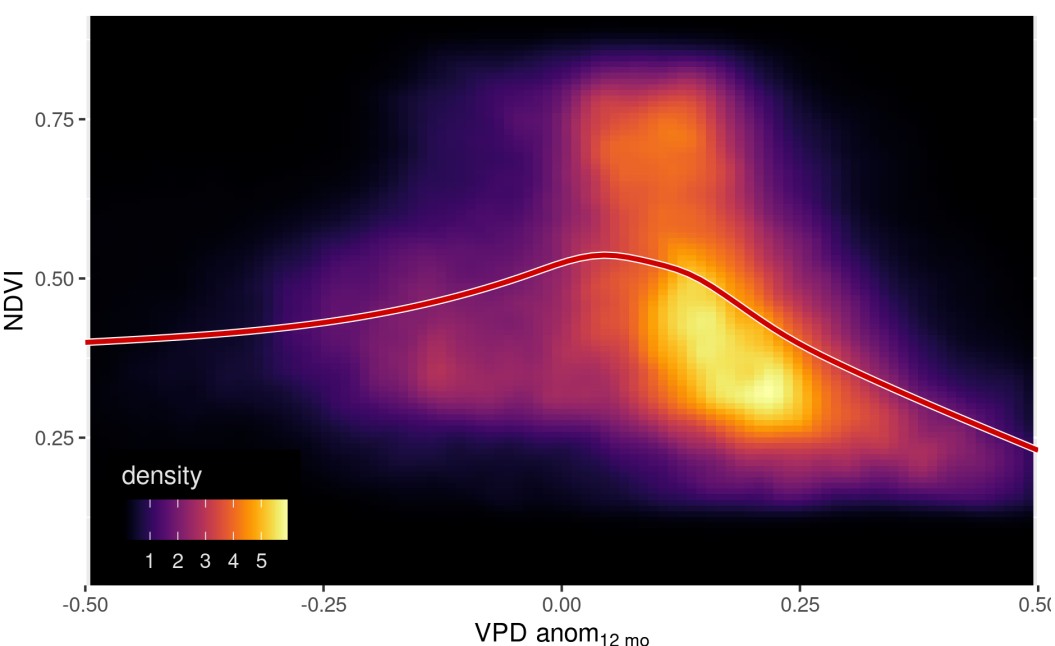

**Figure A11.** The nonlinear dependency of NDVI upon the annual anomaly of vapor pressure deficit. 2D density plot of NDVI and the running 12-month anomaly of vapor pressure deficit. The red line shows a GAM fit.





**Table A1.** Table A1

| Product | Variable | Spatial | Temporal | Reference |
|---|---|---|---|---|
| NOAA-CDR v5 AVHRR surface reflectance | surface reflectance | 0.05° | daily | Vermote et al., 2014 |
| MODIS MCD43 | surface reflectance | 500 m | daily | Schaaf and Wang 2015 |
| MODIS MOD44B | Tree cover, non-tree vegetation cover, non-vegetation cover | 250 m | annual | Dimiceli et al., 2015 |
| MODIS MCD64A1 | Burn Area | 500 m | monthly | Giglio et al., 2015 |
| MODIS MCD15A3H | Leaf Area Index | 500 m | 4-day | Myneni et al., 2015 |
| NVIS v5.1 | Major vegetation classes | 100 m | - | https://www.environment.gov.au/land/native-vegetation/national-vegetation-information-system |
| AWAP Climate | Precip, PET, Tmin, Tmax, Vapor pressure | 0.05° | daily | Jones et al., 2009 |
| ERA5-Land | PET | 9 km | hourly | Copernicus Climate Change Service (C3S) (2017) |
| BoM Major Köppen Climate | Major Köppen Climate zones | - | - | http://www.bom.gov.au/jsp/ncc/climate_averages/climate-classifications |



*Author contributions.* S.W.R., M.G.D.K., P.M., L.A.C., B.E.M., and A.J.P. designed the study. S.W.R. analyzed the data and produced the figures. S.W.R and A.M.U. processed climate data. S.W.R. wrote the first draft with M.G.D.K., and all authors have contributed to writing and revising the manuscript.

*Competing interests.* The authors declare no competing interests.

*Disclaimer.* NA.

*Acknowledgements.* S.W.R., M.G.D.K., A.J.P., L.A.C, and P.M., acknowledge support from the Australian Research Council Discovery Grant (DP190101823). S.W.R., M.G.D.K., A.M.U., and A.J.P. acknowledge support from the ARC Centre of Excellence for Climate Extremes (CE170100023). M.G.D.K. was also supported by the NSW Research Attraction and Acceleration Program and A.M.U. by an ARC Discovery Early Career Researcher Award (DE200100086). B.E.M. acknowledges support from ARC Laureate Fellowship FL190100003.
We are grateful to Randall J. Donohue (CSIRO Land and Water) for valuable discussion and comments on drafts of this manuscript.





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
