# Peer review of "Thirty-eight years of CO2 fertilization have outpaced growing aridity to drive greening of Australian woody ecosystems"

_Biogeosciences, 2021_

## Referee Comment (RC2)

**Review of "Thirty-eight years of $CO_2$ fertilization have outpaced growing aridity to drive greening of Australian woody ecosystems" by Rifai et al. (bg-2021-218)**

In their study, Rifai et al. analyze multi-decadal records of satellite remote sensing of normalized difference vegetation index (NDVI) and examine what role rising $CO_2$ is playing in driving the observed greening. The authors focus their analyses on eastern Australia's woody ecosystems and apply various (linear and non-linear) statistical methods to disentangle the key drivers of vegetation changes. Rifai et al. find that rising atmospheric $CO_2$ contributed to 11.7% increase in NDVI from 1982 to 2019 due to an increase in water-use efficiency, outbalancing the browning-inducing increase in aridity.

Overall, this study provides a very interesting perspective on the driver attribution of vegetation greening trends by examining statistical tools. The result that $CO_2$ fertilization strongly drives greening, which outpaces potential browning due to increasing aridity, is intriguing. I consider the applied statistical methods and the presented results to be robust, and the conclusions drawn to be accurate. The manuscript is very well-written and structured in a clear manner. I have a few general critical points and a rather long list of specific comments.

I recommend a minor revision of the manuscript before publication.

**1 General Comments:**

1.1 *You are applying various statistical methods to evaluate the effect of rising $CO_2$ on the NDVI trend in this study. However, it appears that you are not discussing each method to the same extend. For example, first you focus on the Weibull function (Fig. 5) and later on generalized additive models (Fig. 6 and 7). It does not become clear to the reader when and why you choose the specific models when discussing your findings. Please elaborate why you chose all these statistical models in your research in the first place and why you focus only on specific ones when discussing certain results. Otherwise, why don't you compare the results of all the statistical models you use?*

1.2 *The title and conclusion suggest that rising $CO_2$ could mitigate the adverse effect of increasing aridity on vegetation. Through the whole paper you thus make the assumption that increasing aridity leads to browning. I think you should show explicitly that this assumption holds. You used several statistical models to quantify the effect of $CO_2$, so you could reconstruct NDVI patterns without it and see how Australian woody ecosystems would look like without $CO_2$ fertilization.*

1.3 *The WUE model used here might be a too simplified representation of the studied processes. First, it focuses on atmospheric dryness and ignores soil dryness, i.e. soil moisture is not taken into account in this model. Second, the assumption that all $CO_2$-induced greening is due to increased WUE is also limited. Part of the vegetation greening could also be driven by increased productivity due to the RuBisCO-machinery response to $CO_2$ (Walker et al., 2021). Can you elaborate why your theoretical WUE model*

*is sufficient to explain the observed changes?*

**2 Specific Comments:**

*2.1 Please stick to the tenses, i.e. do not switch between present and past tense when describing your results. I recommend that you always use present tense when talking about your study, i.e. when describing your methods, your results etc., and use past tense when referring to already published studies.*

*2.2 You have some issues with your BibLaTeX / BibTeX setup. Sometimes brackets are missing, sometimes there are too many. Sometimes the "year" is missing in your citation (e.g. L25), or the BibTeX key does not work at all (e.g. L260).*

*2.3 L32: You show different seasons in Fig. A2, which you don't discuss here. Better to refer to Fig. 2 here, right?*

*2.4 L40: Don't understand why you refer to Zhu et al. (2016) here. They did not show that the attribution is challenging, they rather reported successful attribution results and state that most of the "greening" can be tied to $CO_2$ fertilization. A recently published study by Winkler et al. (2021), for example, showed that the attribution of $CO_2$ effect is challenging in observations / remote sensing.*

*2.5 L43-44: You write "The greening trend is caused by increased leaf area, which has resulted from increased atmospheric $CO_2$ concentrations". This statement reads as an absolute finding. In the sentence before, however, you write that the attribution is challenging. Can you clarify?*

*2.6 L54: Cortés et al. (2021) also show that testing for significance crucially depends on subjectively chosen criteria. One can set up a significance test such that no grid cell exhibits a significant trend, and probably the opposite: all trends are significant! I'm just wondering where we advance our scientific understanding there. I think the question is whether an observed trend, significant or not, is consistent with our expectations based on our understanding of the processes, or is it plausible in terms of our understanding of the system? If not, are we missing something? If yes, let's investigate what could be driving the trend! This is less a comment to your study, but overall to the discipline of studying greening trends and others.*

*2.7 L56: Another aspect is whether modeled LAI and observed LAI is really the same thing, conceptually.*

*2.8 L66-70: Here, you should include the fact that you approach the problem with different statistical methods.*

*2.9 L85: Have you also used other estimates, daily mean, daily max value and checked the sensitivity of your methods to this decision?*

*2.10 L95: What Organization? Please check all citations and references throughout the paper!*

*2.11 L95: Please specify the acronym MI.*

*2.12 L102: 0.05° is quite a high resolution for the AVHRR reflectances. Can you shortly note here, how this was made possible?*

2.13 *L127: Please follow the guidelines for math notations:* [https://publications.copernicus.org/for_authors/manuscript_preparation.html#math](https://publications.copernicus.org/for_authors/manuscript_preparation.html#math)

2.14 **2.4 Estimating contribution of CO2 and climate toward NDVI trends***: Can you be more precise and say how many approaches to statistical modelling you implemented here, and why you chose these? It seems you have fitted 5 different stats. models? If so, why are you not discussing all of them to the same extent?*

2.15 *L150: You write "The relationship between NDVI and the running 12-month mean of P:PET* **was** *strongly nonlinear..."; shouldn't it rather read "***is** *nonlinear", right? Please see also comment 2.1.*

2.16 *L155: Weibull function should refer to Fig. 5 instead of Fig. 4, right?*

2.17 *L156: "We focus on the Weibull..." Are the Weibull models fitted for each grid cell? Please specify in the method section.*

2.18 *L186 & L189: ":" before instead of "." after "follows"*

2.19 *L260: You mix a bit the Results and Discussion section, e.g. in the Results sub-section "***3.2 Empirical attribution of the CO$_2$ effect***" you already discuss other studies and put your results in perspective. That should be part of the Discussion sections. Either you clearly separate Results and Discussion, or you merge them completely to one section.*

2.20 *L304: It might be interesting to discuss also the results of a recent study by Winkler et al. (2021). They approached the problem of CO$_2$ fertilization attribution using a different approach based on causal theory, also looking into Australian vegetated lands to some extent.*

2.21 *L345: Maybe this is because you did not take into account the RuBisCO response to CO$_2$, which is taken into account in biosphere models (Walker et al., 2021).*

2.22 *L347-349: Maybe another limiting factor such as nutrients could kick in.*

2.23 *L383: "Some may question the veracity... ". This formulation is a bit vague: are there some studies that indicate that the veracity is limited?*

2.24 *Equation 7: Are the variables (e.g. CO$_2$) in the linear model normalized or not? It's important to know to what extent the coefficients of the different predictor variables are comparable. Please specify.*

2.25 *L195: A is leaf level carbon assimilation...*

2.26 *Equation 11: Is D the same term as VPD in other equations? Please use consistent abbreviation.*

2.27 *Equation 13: no definition for parameter "L"*

2.28 *Equation 16: no definition for parameter "F"*

2.29 *L290: a missing closing bracket*

2.30 *L317: There is no Fig. 6(a).*

2.31 *Figure 3: Why not show AVHRR record for the second time period?*

2.32 *Figure 5 (a): No explanation of the calculation of $CO_2$ and $\Delta NDVI$ (%) in the method section.*

2.33 *Figure 5, caption: "… are plotted in gray for panels B and C" should be "panels (c) and (d), right?*

2.34 *Figure 6: Can you explain why some models predict a mean negative change in NDVI with rising $CO_2$ – is this plausible, or is this only due to stats. uncertainty. If the latter is true, how does that generally impact on the reliability of your estimates?*

2.35 *Figure 6: Why are you only comparing two different methods here? As I understand reading your methods section, you are using more than two stats. modeling approaches.*

2.36 *Figure 7: Can you show predictions for NDVI without $CO_2$ – it's interesting to see the effect of intensifying aridity in the absence of $CO_2$ fertilization; are the ecosystems browning then?*

2.37 *Figure 7: Can you be more explicit about the labels: Does "Tropical" mean "Tropical Forests"? What is Temperate Tas.?*

2.38 *Figure 7 caption: Please use the LaTeX math mode correctly. Math-mode is always putting the text in itatlic, but this is not correct most of the time. This introduces also some inconsistencies in your manuscript, e.g. you put NDVI in non-itatlic but $\Delta NDVI$ in italic. Also units and chemical formulas like $CO_2$ etc. should not be typeset in italic! Please see also comment 2.12.*

2.39 *Figure 7: Are the captions missing for panels (d) and (e)?*

2.40 *Figure 8a: Can you provide some estimate of uncertainty for the different regression lines, e.g. the standard error? How robust are these linear trends?*

2.41 *Figure 8c: I don't get your biome definition really. What is "Tree Vegetation" in "Grassland"? Are you showing the shifts from non-woody to woody species composition here?*

2.42 *Figure 8 (c): "The 25, 50 and 75% quantiles are overlaid." However, in some distributions, there are only 1 or 2 lines.*

2.43 *Figure A9: According to L214, your theoretical assumption only holds as long as $LAI <= 3$. The values in x-axis in Fig. A9, however, exceed 4. How does that impact your analysis here?*

2.44 *Figure A2: This figure could be extended with another panel for P/PET.*

2.45 *Figure 3. One idea could be to move Fig. 3(a) to Fig. 2., and add another supplementary figure instead. This one could include Fig 3 (b), (c) and the merged NDVI between 1982-2019 for each season.*

**References**

Cortés, J., Mahecha, M. D., Reichstein, M., Myneni, R. B., Chen, C., and Brenning, A. (2021). Where Are Global Vegetation Greening and Browning Trends Significant? *Geophysical Research Letters*, 48(6):e2020GL091496.

Walker, A. P., Kauwe, M. G. D., Bastos, A., Belmecheri, S., Georgiou, K., Keeling, R. F., McMahon, S. M., Medlyn, B. E., Moore, D. J. P., Norby, R. J., Zaehle, S., Anderson-Teixeira, K. J., Battipaglia, G., Brienen, R. J. W., Cabugao, K. G., Cailleret, M., Campbell, E., Canadell, J. G., Ciais, P., Craig, M. E., Ellsworth, D. S., Farquhar, G. D., Fatichi, S., Fisher, J. B., Frank, D. C., Graven, H., Gu, L., Haverd, V., Heilman, K., Heimann, M., Hungate, B. A., Iversen, C. M., Joos, F., Jiang, M., Keenan, T. F., Knauer, J., Körner, C., Leshyk, V. O., Leuzinger, S., Liu, Y., MacBean, N., Malhi, Y., McVicar, T. R., Penuelas, J., Pongratz, J., Powell, A. S., Riutta, T., Sabot, M. E. B., Schleucher, J., Sitch, S., Smith, W. K., Sulman, B., Taylor, B., Terrer, C., Torn, M. S., Treseder, K. K., Trugman, A. T., Trumbore, S. E., van Mantgem, P. J., Voelker, S. L., Whelan, M. E., and Zuidema, P. A. (2021). Integrating the evidence for a terrestrial carbon sink caused by increasing atmospheric CO2. *New Phytologist*, 229(5):2413–2445.

Winkler, A. J., Myneni, R. B., Hannart, A., Sitch, S., Haverd, V., Lombardozzi, D., Arora, V. K., Pongratz, J., Nabel, J. E. M. S., Goll, D. S., Kato, E., Tian, H., Arneth, A., Friedlingstein, P., Jain, A. K., Zaehle, S., and Brovkin, V. (2021). Slowdown of the greening trend in natural vegetation with further rise in atmospheric $CO_2$. *Biogeosciences*, 18(17):4985–5010.

Zhu, Z., Piao, S., Myneni, R. B., Huang, M., Zeng, Z., Canadell, J. G., Ciais, P., Sitch, S., Friedlingstein, P., Arneth, A., Cao, C., Cheng, L., Kato, E., Koven, C., Li, Y., Lian, X., Liu, Y., Liu, R., Mao, J., Pan, Y., Peng, S., Peñuelas, J., Poulter, B., Pugh, T. A. M., Stocker, B. D., Viovy, N., Wang, X., Wang, Y., Xiao, Z., Yang, H., Zaehle, S., and Zeng, N. (2016). Greening of the Earth and its drivers. *Nature Climate Change*, 6(8):791–795.

---

## Author Comment (AC1)

**Referee 1 comments:**

This is an important paper and is a novel and interesting analysis of the magnitude of the CO2 fertilisation effect on changes to green cover despite drought, fire and LUC across the forests of eastern Australia. It was well written and uses a statistical approach (as opposed to a ecosystem model) to separate the relative impact of drought on the CO2 fertilisation. Clever analysis, novel presentation of results (eg Figure 1) and was very well written.

The only issue I had was with the coarse resolution remote sensing product that was used to determine cover change.

L50 The Landsat based green vegetation fractional cover product (GV) is potentially superior to NDVI and I am wondering why this was not used. Monthly data are available from GeoSci Australia's data cube at 25 m resolution and is derived from Landsat 5 TM, ETM and OLI from 1987 onwards and are geometrically corrected, converted to surface reflectance, adjusted for solar illumination and viewing angles and masked for cloud and cloud shadow (see Lewis et al 2017, Gill et al 2017). The data are 'analysis ready' and this product is likely too be superior to NDVI and includes fractional bare soil and non-photosynthetic cover ideal for the analysis of drought impacts. Earlier imagery could be used to get back to 1982.

The fractional cover is at a far higher resolution (~25 m) and is a sub-pixel fraction and provides a finer-grained information than NDVI. NDVI is sensitive to saturation at the high cover, and is sensitive to variation in vegetation structure (i.e., bare ground), differences in canopy openness and complex seasonality associated with overstorey and understorey veg components. The fractional cover product also avoids dealing with both the AVHRR NDVI and MODIS NDVI data.

*\* We used the coarser resolution product of AVHRR because it was a closer match to the spatial resolution of the AWAP climate data (~0.05°). It is also unclear how the fractional cover would respond to closed-canopy forest where the fractional cover might be expected to be saturated. We caution that there might be several separate issues regarding differences and degradations with the Landsat multispectral sensors. For example, Landsat 7's ETM+ had a well known malfunction that caused striping, Landsat 5, 7, and 8 have different radiometric resolution, and Landsat 5's TM sensor experienced decay and the platform's solar zenith angle increased during the later years of Landsat 5's 25+ year mission. This opens up various unknowns, and while this would be an interesting dimension to the question of the $CO_2$ effect, it would necessitate a separate piece of work.*

The Discussion was excellent and drew the threads together well, and highlighted the notion that examining response of Australian vegetation will be helpful for other assessing impacts from other vegetation systems globally. While Australian focused, the approach and outcomes is of global significance.

**So accept subject to consideration of using fractional cover. The next study could be useful to look at potential decrease in understory vegetation as overstory thickens. This has implication for prodciutvity and biodiversity.**

*\* We agree the Geosci Australia fractional cover product would be ideal for future studies and we acknowledge the fractional cover would be a useful way to more resolutely identify shifts in vegetation cover. However we have not provided the requested analysis because the fractional cover data is more than 160x times the resolution, very large (100+ GB), and well beyond our capacity to process in due time for meeting the response deadline of the journal. Moreover this work is part of the lead author's postdoc for the last two years, and unfortunately cannot allocated the months necessary to re-do the analysis with the Geosci Australia Fractional Cover product because he is starting a new postdoc on a different topic.*

*We acknowledge the utility of fractional cover in the discussion for 'future work' in section 4.6 Vegetation composition shifts, with the following text:*
*"Future work may seek to uncover trends in the green vegetation fraction by analyzing higher resolution data derived from the Landsat constellation, such as Geoscience Australia's vegetation fractional cover product (Gill et al., 2017)."*

---

## Author Comment (AC2)

**Referee 2 comments:**

In their study, Rifai et al. analyze multi-decadal records of satellite remote sensing of normalized difference vegetation index (NDVI) and examine what role rising CO2 is playing in driving the observed greening. The authors focus their analyses on eastern Australia's woody ecosystems and apply various (linear and non-linear) statistical methods to disentangle the key drivers of vegetation changes. Rifai et al. find that rising atmospheric CO2 contributed to 11.7% increase in NDVI from 1982 to 2019 due to an increase in water-use efficiency, outbalancing the browning-inducing increase in aridity.

Overall, this study provides a very interesting perspective on the driver attribution of vegetation greening trends by examining statistical tools. The result that CO2 fertilization strongly drives greening, which outpaces potential browning due to increasing aridity, is intriguing. I consider the applied statistical methods and the presented results to be robust, and the conclusions drawn to be accurate. The manuscript is very well-written and structured in a clear manner. I have a few general critical points and a rather long list of specific comments.

*We thank the reviewer for their positive evaluation of our manuscript.*

I recommend a minor revision of the manuscript before publication.

**General Comments:**
**1.1 You are applying various statistical methods to evaluate the effect of rising CO2 on the NDVI trend in this study. However, it appears that you are not discussing each method to the same extend. For example, first you focus on the Weibull function (Fig. 5) and later on generalized additive models (Fig. 6 and 7). It does not become clear to the reader when and why you choose the specific models when discussing your findings. Please elaborate why you chose all these statistical models in your research in the first place and why you focus only on specific ones when discussing certain results. Otherwise, why don't you compare the results of all the statistical models you use?**
*\* In methods section 2.4, we have added an opening sentence to each paragraph that introduces a new model in order to explain the reason for including the model form in the analysis.*

**1.2 The title and conclusion suggest that rising CO2 could mitigate the adverse effect of increasing aridity on vegetation. Through the whole paper you thus make the assumption that increasing aridity leads to browning. I think you should show explicitly that this assumption holds. You used several statistical models to quantify the effect of CO2 , so you could reconstruct NDVI patterns without it and see how Australian woody ecosystems would look like without CO2 fertilization.**
*\* Figure 1 attempts to convey that NDVI follows a broad saturating function of P:PET. The effect of anomalously high VPD causing a decline in NDVI is shown explicitly in Figure A11. Despite the decrease in P:PET, we see that NDVI has largely increased - hence the concentration on*

*the CO2 effect as the most plausible mechanism. In* equation 18, the WUE model shows the expected relative change in NDVI due to relative changes in VPD.

*Ultimately this manuscript is focused on attributing greening to the $CO_2$ effect. We provide a figure (below) showing the counterfactual of the observed changes in aridity but without a change in $CO_2$. We are reluctant to include this in the main text, (1) in order to keep the manuscript focused on the $CO_2$ effect, and (2) because this would essentially add another possibly confusing section and figure to the manuscript.*

**1.3 The WUE model used here might be a too simplified representation of the studied processes. First, it focuses on atmospheric dryness and ignores soil dryness, i.e. soil moisture is not taken into account in this model. Second, the assumption that all CO2 -induced greening is due to increased WUE is also limited. Part of the vegetation greening could also be driven by increased productivity due to the RuBisCO machinery response to CO2 (Walker et al., 2021). Can you elaborate why your theoretical WUE model is sufficient to explain the observed changes?**
*\*We agree with the reviewer that the WUE model is simple. To clarify (see also below), we are in fact testing the assumptions of this model vs our statistical approach, rather than implying the model does have sufficient explanatory power.*

*First, we agree with their initial point about ignoring changes in soil dryness; however, we consider this to be a minor effect given perturbations in soil dryness are typically short-lived (seasonal to annual) relative to the overall trend across 38 years. In fact, there is no adequate way to capture feedbacks due to changes in soil water availability as this is not captured via observations and thus, would require the use of more complicated models that have long standing, systematic biases as water becomes limiting (Ukkola et al., 2016 Environmental Research Letters). In Fig. 5, we statistically constructed a counterfactual dNDVI response to the $CO_2$ increase where there was no change in the precipitation supply.*

*Second, with respect to the reviewer's next point, the changes in the biochemical effect of rising [$CO_2$] on photosynthesis are captured by our model. The model proposed by Donohue et al. 2013 GRL, hypothesised that in water limited environments, the $CO_2$ induced change in WUE is shared **evenly** between the change in assimilation (direct biochemical effect and indirect change via a change in leaf area) and the change in transpiration (a function of the change in LAI: directly via increased growth and indirectly via soil water savings). Donohue et al. did not justify the even split between assimilation and evaporation, but instead tested their prediction against the observed change in fractional cover, finding good support for the predicted change across the globe's warm, arid environments. In our paper, we further tested this assumption with a longer time series and more focussed spatial domain; we did not find strong support for this hypothesis. Thus, our paper makes an important contribution by re-examining the support for the hypothesis and interpretation advanced by Donohue et al.; discussed further in sections 3.3, 4.2, 4.4 and Figure A9. Where we listed reasons for the divergence from the expectations of the simple model (i.e. "These deviations may have resulted from ecosystems processes beyond the scope of a simple model, such as … "), we now add "belowground root growth".*

**Specific Comments:**

**2.1 Please stick to the tenses, i.e. do not switch between present and past tense when describing your results. I recommend that you always use present tense when talking about your study, i.e. when describing your methods, your results etc., and use past tense when referring to already published studies.**

*We have ensured all verbs are in the past tense, except in the situations where it does not make sense such as in the introduction and discussion. Respectfully we disagree with this recommendation because it has long been acceptable to use the past tense in the methods and results because these describe actions and findings that occurred in the past.*

**2.2 You have some issues with your BibLaTeX / BibTeX setup. Sometimes brackets are missing, some times there are too many. Sometimes the "year" is missing in your citation (e.g. L25), or the BibTeX key does not work at all (e.g. L260).**

*corrected; The year was missing because the article was in-press at the time of submission.*

**2.3 L32: You show different seasons in Fig. A2, which you don't discuss here. Better to refer to Fig. 2 here, right?**

*corrected*

**2.4 L40: Don't understand why you refer to Zhu et al. (2016) here. They did not show that the attribution is challenging, they rather reported successful attribution results and state that most of the "greening" can be tied to CO2 fertilization. A recently published study by Winkler et al. (2021), for example, showed that the attribution of CO2 effect is challenging in observations / remote sensing.**

*Corrected, we put the Zhu 2016 reference in the wrong location in the sentence. We highlight the Frankenberg 2021 and Zhu 2021 technical comments with respect to the difficulty of attributing the CO2 effect.*

**2.5 L43-44: You write "The greening trend is caused by increased leaf area, which has resulted from increased atmospheric CO2 concentrations". This statement reads as an absolute finding. In the sentence before, however, you write that the attribution is challenging. Can you clarify?**

*Corrected, we inserted the word 'likely' with respect to $CO_2$ driven greening.*

**2.6 L54: Cortés et al. (2021) also show that testing for significance crucially depends on subjectively chosen criteria. One can set up a significance test such that no grid cell exhibits a significant trend, and probably the opposite: all trends are significant! I'm just wondering where we advance our scientific understanding there. I think the question is whether an observed trend, significant or not, is consistent with our expectations based on our understanding of the processes, or is it plausible in terms of our understanding of the system? If not, are we missing something? If yes, let's investigate what could be driving the trend! This is less a comment to your study, but overall to the discipline of studying greening trends and others.**

*We understand this is a comment, so we did not amend the text in response. We argue that the Cortés 2021 definition of significance is based on their own derivation of permutation tests and clustering - which are not necessarily objective. They offer another definition and method for 'significance'. For example, Ives et al., 2021 offer their own methodology and definition of 'significance' for greening too (https://doi.org/10.1016/j.rse.2021.112678). Problems with Frequentist p-value 'significance' are well discussed in the literature so we do not address them here. More than 90% of the pixel locations showed a positive greening trend, and effectively all locations showed a positive effect from $CO_2$. We focused on effect sizes which is what is biologically relevant. Moreover we demonstrated the statistically isolated effects of $CO_2$ on greening are consistent with (an independently derived) theoretical expectation from WUE. We argue this manuscript is focused on identifying what is driving the trend: (1) $CO_2$, (2) changes in aridity, either due to VPD or changes in precipitation, and (3) disturbance.*

**2.7 L56: Another aspect is whether modeled LAI and observed LAI is really the same thing, conceptually.**
*We agree that modeled LAI is merely an estimate, dependent upon algorithmic considerations which are different between the AVHRR and MODIS LAI products. The discrepancy of observed LAI with the MODIS LAI product is well documented over Australian ecosystems (Leuning et al., 2005 Agriculture and Forest Meteorology; Hill et al., 2006 Remote Sensing of Environment). Again, this is part of why we did not use RS-modeled LAI in this analysis.*

**2.8 L66-70: Here, you should include the fact that you approach the problem with different statistical methods.**
*done*

**2.9 L85: Have you also used other estimates, daily mean, daily max value and checked the sensitivity of your methods to this decision?**
*No because there is not an obvious physiological relevance of night-time VPD on WUE.*

**2.10 L95: What Organization? Please check all citations and references throughout the paper!**
*corrected*

**2.11 L95: Please specify the acronym MI.**
*done*

**2.12 L102: 0.05° is quite a high resolution for the AVHRR reflectances. Can you shortly note here, how this was made possible?**
*This is the native resolution of the NOAA CDR AVHRR: Surface Reflectance, Version 5 product (Eric Vermote, Chris Justice, Ivan Csiszar, Jeff Eidenshink, Ranga Myneni, Frederic Baret, Ed Masuoka, Robert Wolfe, Martin Claverie and NOAA CDR Program (2014): NOAA Climate Data Record (CDR) of AVHRR Surface Reflectance, Version 4. NOAA National Climatic Data Center.).*
*Also see:*
*(https://www.ncei.noaa.gov/access/metadata/landing-page/bin/iso?id=gov.noaa.ncdc:C01557)*

**2.13 L127: Please follow the guidelines for math notations: https:// publications.copernicus.org/ for authors/ manuscript preparation.html#math**
*We are unclear where the violation is of math notation guidelines for equation 1.*

**2.14 2.4 Estimating contribution of CO2 and climate toward NDVI trends: Can you be more precise and say how many approaches to statistical modelling you implemented here, and why you chose these? It seems you have fitted 5 different stats. models? If so, why are you not discussing all of them to the same extent?**
*Noted, we amended the text to mention the six methods.*

**2.15 L150: You write "The relationship between NDVI and the running 12-month mean of P:PET was strongly nonlinear..."; shouldn't it rather read "is nonlinear", right? Please see also comment 2.1.**
*We removed the word 'strongly'.*

**2.16 L155: Weibull function should refer to Fig. 5 instead of Fig. 4, right?**
*corrected*

**2.17 L156: "We focus on the Weibull. . . " Are the Weibull models fitted for each grid cell? Please specify in the method section.**
*done*

**2.18 L186 & L189: ":" before instead of "." after "follows"**
*done*

**2.19 L260: You mix a bit the Results and Discussion section, e.g. in the Results sub-section "3.2 Empirical attribution of the CO2 effect" you already discuss other studies and put your results in perspective. That should be part of the Discussion sections. Either you clearly separate Results and Discussion, or you merge them completely to one section.**
* Here we disagree and argue that citing the Wang 2020 study here is important to note the context of the result. It is discussed further in the Discussion.*

**2.20 L304: It might be interesting to discuss also the results of a recent study by Winkler et al. (2021). They approached the problem of CO2 fertilization attribution using a different approach based on causal theory, also looking into Australian vegetated lands to some extent.**
* Thank you, we now cite Winkler et al. (2021).*

**2.21 L345: Maybe this is because you did not take into account the RuBisCO response to CO2 , which is taken into account in biosphere models (Walker et al., 2021).**
* Please see responses to 1.3.*

**2.22 L347-349: Maybe another limiting factor such as nutrients could kick in.**
*\* We have amended the text to incorporate the suggestion as:*
*"These deviations may have resulted from ecosystems processes beyond the scope of a simple model, such as more severe nutrient limitations, the phenology of the vegetation composition, and disturbances not captured by the satellite products (e.g. small fires, grazing). Browning in the Grassland region (Fig. 3b,A3)"*

**2.23 L383: "Some may question the veracity... ". This formulation is a bit vague: are there some studies that indicate that the veracity is limited?**
*\* We have replaced the word 'veracity' with 'accuracy' and cited two studies that examine the relevant accuracy of the VCF product in forested to savanna ecosystems.*

**2.24 Equation 7: Are the variables (e.g. CO2 ) in the linear model normalized or not? It's important to know to what extent the coefficients of the different predictor variables are comparable. Please specify.**
*\*Yes, they are 'z-score' transformed, mentioned in the text following equation 7.*

**2.25 L195: A is leaf level carbon assimilation. . .**
*\*corrected*

**2.26 Equation 11: Is D the same term as VPD in other equations? Please use consistent abbreviation.**
*\*Noted, we use D only for describing the WUE model as this has been the convention in the leaf-level analytical model literature (e.g. Medlyn et al., 2011; Donohue et al., 2013).*

**2.27 Equation 13: no definition for parameter "L"**
*\*corrected*

**2.28 Equation 16: no definition for parameter "F"**
*\* This was defined as foliar cover, after equation 15 on (previous) L214.*

**2.29 L290: a missing closing bracket**
**\****corrected*

**2.30 L317: There is no Fig. 6(a).**
*\*corrected*

**2.31 Figure 3: Why not show AVHRR record for the second time period?**
*\* Here the MODIS NDVI record is objectively better because the AVHRR record begins to deteriorate in 2016 due to a decline in solar zenith angle.*

**2.32 Figure 5 (a): No explanation of the calculation of CO2 and ∆NDVI (%) in the method section.**

*\* The figure legend text reads: "Panel (a) maps the total predicted contribution of $CO_2$ towards the relative increase of NDVI between 1982-2019 assuming no anomaly of P:PET."*
*Here we used the same model $CO_2$ that corresponds to the Mauna Loa trend record.*

**2.33 Figure 5, caption: ". . . are plotted in gray for panels B and C" should be "panels (c) and (d), right?**
*\* corrected*

**2.34 Figure 6: Can you explain why some models predict a mean negative change in NDVI with rising CO2 – is this plausible, or is this only due to stats. uncertainty. If the latter is true, how does that generally impact on the reliability of your estimates?**
*\* All statistical model estimates have uncertainty whether they are biophysically possible or not. Note that these are a distribution of several thousand pixel location fits of a robust linear model. We argue it is unreasonable to assume that every pixel level fit will be capable of determining the sign of the $CO_2$ effect, especially when the effect size is as small as it is in the Arid region.*

**2.35 Figure 6: Why are you only comparing two different methods here? As I understand reading your methods section, you are using more than two stats. modeling approaches.**
*\* We chose to only display two different approaches, a pixel-level specific method (the RLM), and the cross-pixels GAM method. This already results in six bars per Köppen climate zone, so we felt adding more bars would make this figure overwhelming. Also, the Weibull function fits are already shown in Figure 5b so that might be redundant to include them again in Figure 6. , we have added text to the figure legend to clarify the different modeling approaches.*

**2.36 Figure 7: Can you show predictions for NDVI without CO2 – it's interesting to see the effect of intensifying aridity in the absence of CO2 fertilization; are the ecosystems browning then?**
*\* Browning is indeed another interesting process, but it was not the focus of this study. We provide the requested figure here, but have not added it to the main text to keep the manuscript focused on the question of the $CO_2$ contribution to greening.*

[Figure]

Counterfactual Relative Difference in NDVI: 1982-2019

**2.37 Figure 7: Can you be more explicit about the labels: Does "Tropical" mean "Tropical Forests"? What is Temperate Tas.?**

*\* The methods section "Climate and remote sensing data sets" and Figure 2b explains these are simplified Köppen climate zones used by the Australian Bureau of Meteorology. We have added "Köppen Climate Zone" to the x-axis label of figure 7 to further clarify.*

**2.38 Figure 7 caption: Please use the LaTeX math mode correctly. Math-mode is always putting the text in itatlic, but this is not correct most of the time. This introduces also**

**some inconsistencies in your manuscript, e.g. you put NDVI in non-itatlic but ∆NDVI in italic. Also units and chemical formulas like CO2 etc. should not be typeset in italic! Please see also comment 2.12.**

*We used the Rmarkdown template suggested by Copernicus (https://publications.copernicus.org/for_authors/manuscript_preparation.html#templates) and have not changed the template markdown settings. We have changed $CO_2$ to $C_a$ in all the equations. Otherwise we have tried to fix dNDVI, but this is beyond our knowledge of the inner workings of the Rmarkdown format and template so we believe this will get fixed by the copy editor.*

**2.39 Figure 7: Are the captions missing for panels (d) and (e)?**
*corrected*

**2.40 Figure 8a: Can you provide some estimate of uncertainty for the different regression lines, e.g. the standard error? How robust are these linear trends?**
*The trends actually do show a barely visible uncertainty interval. The estimates and standard error for each trend has been added to the figure legend.*

**2.41 Figure 8c: I don't get your biome definition really. What is "Tree Vegetation" in "Grassland"? Are you showing the shifts from non-woody to woody species composition here?**
*These are standard climate zone definitions derived from a simplified Köppen framework that is commonly used by the Australian Bureau of Meteorology (Fig. 1b; Methods section 2.2). Despite being named the 'Grassland', there are some woody ecosystems in this zone.*

**2.42 Figure 8 (c): "The 25, 50 and 75% quantiles are overlaid." However, in some distributions, there are only 1 or 2 lines.**
* *Changed to only show the median.*

**2.43 Figure A9: According to L214, your theoretical assumption only holds as long as LAI <= 3. The values in x-axis in Fig. A9, however, exceed 4. How does that impact your analysis here?**
*The theoretical WUE model is very simple with approximations. MODIS modelled LAI is well known to be biased high in Australia compared to field observations, but still only exceeds 3 in a relatively small proportion of pixel locations. Further discussion into the scaling of transpiration with LAI > 3 is beyond the scope of our analysis.*

**2.44 Figure A2: This figure could be extended with another panel for P/PET.**
* *The change in annual P:PET is already shown in Figure 2. Given the seasonal changes in P and PET are shown in Fig. A2, it is not clear to us what additional value a seasonal P:PET plot would show.*

**2.45 Figure 3. One idea could be to move Fig. 3(a) to Fig. 2., and add another supplementary figure instead. This one could include Fig 3 (b), (c) and the merged NDVI between 1982-2019 for each season.**

*\*It is not specified what advantage this would bring, therefore we respectfully disagree with this suggestion because it would add another supplementary figure and disrupt the order statements in the text.*

**References**

**Cortés, J., Mahecha, M. D., Reichstein, M., Myneni, R. B., Chen, C., and Brenning, A. (2021). Where Are Global Vegetation Greening and Browning Trends Significant? Geophysical Research Letters, 48(6):e2020GL091496.**

**Walker, A. P., Kauwe, M. G. D., Bastos, A., Belmecheri, S., Georgiou, K., Keeling, R. F., McMahon, S. M., Medlyn, B. E., Moore, D. J. P., Norby, R. J., Zaehle, S., Anderson-Teixeira, K. J., Battipaglia, G., Brienen, R. J. W., Cabugao, K. G., Cailleret, M., Campbell, E., Canadell, J. G., Ciais, P., Craig, M. E., Ellsworth, D. S., Farquhar, G. D., Fatichi, S., Fisher, J. B., Frank, D. C., Graven, H., Gu, L., Haverd, V., Heilman, K., Heimann, M., Hungate, B. A., Iversen, C. M., Joos, F., Jiang, M., Keenan, T. F., Knauer, J., Körner, C., Leshyk, V. O., Leuzinger, S., Liu, Y., MacBean, N., Malhi, Y., McVicar, T. R., Penuelas, J., Pongratz, J., Powell, A. S., Riutta, T., Sabot, M. E. B., Schleucher, J., Sitch, S., Smith, W. K., Sulman, B., Taylor, B., Terrer, C., Torn, M. S., Treseder, K. K., Trugman, A. T., Trumbore, S. E., van Mantgem, P. J., Voelker, S. L., Whelan, M. E., and Zuidema, P. A. (2021). Integrating the evidence for a terrestrial carbon sink caused by increasing atmospheric CO2. New Phytologist, 229(5):2413–2445.**

**Winkler, A. J., Myneni, R. B., Hannart, A., Sitch, S., Haverd, V., Lombardozzi, D., Arora, V. K., Pongratz, J., Nabel, J. E. M. S., Goll, D. S., Kato, E., Tian, H., Arneth, A., Friedlingstein, P., Jain, A. K., Zaehle, S., and Brovkin, V. (2021). Slowdown of the greening trend in natural vegetation with further rise in atmospheric CO2. Biogeosciences, 18(17):4985–5010.**

**Zhu, Z., Piao, S., Myneni, R. B., Huang, M., Zeng, Z., Canadell, J. G., Ciais, P., Sitch, S., Friedlingstein, P., Arneth, A., Cao, C., Cheng, L., Kato, E., Koven, C., Li, Y., Lian, X., Liu, Y., Liu, R., Mao, J., Pan, Y., Peng, S., Peñuelas, J., Poulter, B., Pugh, T. A. M., Stocker, B. D., Viovy, N., Wang, X., Wang, Y., Xiao, Z., Yang, H., Zaehle, S., and Zeng, N. (2016). Greening of the Earth and its drivers. Nature Climate Change, 6(8):791–795.**